# A reprogrammable mechanical metamaterial with origami functional-group transformation and ring reconfiguration

Xinyu Hu [1], Ting Tan[2], Benlong Wang [1] & Zhimiao Yan [1] ✉

Recent advancements in reprogrammable metamaterials have enabled the development of intelligent matters with variable special properties in situ. These metamaterials employ intra-element physical reconfiguration and inter-element structural transformation. However, existing mono-characteristic homo-element mechanical metamaterials have limited reprogramming functions. Here, we introduce a reprogrammable mechanical metamaterial composed of origami elements with heterogeneous mechanical properties, which achieves various mechanical behavior patterns by functional group transformations and ring reconfigurations. Through the anisotropic assembly of two heterogeneous elements into a functional group, we enable mechanical behavior switching between positive and negative stiffness. The resulting polygonal ring exhibits rotational deformation, zero Poisson's ratio stretching/compression deformation, and negative Poisson's ratio auxetic deformation. Arranging these rings periodically yields homogeneous metamaterials. The reconfiguration of quadrilateral rings allows for continuous fine-tunability of the mechanical response and negative Poisson's ratio. This mechanical metamaterial could provide a versatile material platform for reprogrammable mechanical computing, multi-purpose robots, transformable vehicles and architectures at different scales.

Mechanical metamaterials represent a category of innovative materials that, by designing specific physical elements, surpass the constraints of certain natural laws and achieve unique mechanical properties[1–4], such as alternating Poisson's ratio[5,6], multi-stability[7,8], negative compressibility[9,10], chirality[11,12], and tunable stiffness[13,14]. To realize these properties, precise design and implementation of metamaterials are necessary. However, the evolving functional requirements pose challenges for reprogrammable metamaterials, which should allow for on-demand multi-function regulation.

One approach towards achieving reprogrammable metamaterials involves the reversible reconfiguration of intra-elements[15–17]. For instance, the switching between stable states in a bistable element enables the creation of a mechanical bit for stable memory. On the other hand, reconfiguration at the inter-element level facilitates the achievement of reprogrammable multifunctionality[18]. For example, the structural transformation of inter-elements in modular kirigami metamaterial enables different mechanical responses under reprogrammable shearing and compression deformations. However, these reprogrammable metamaterials based on homo-elements have limited functions due to their mono-characteristics at the unit cell level. Therefore, diverse research on deformation modes and mechanical responses of unit cells, as well as their mixed use and rational connection design, are crucial in realizing the transformable and reconfigurable properties required for reprogrammable multifunctional applications.

[1]State Key Laboratory of Ocean Engineering, Department of Mechanics, School of Naval Architecture, Ocean & Civil Engineering, Shanghai Jiao Tong University, 200240 Shanghai, China. [2] State Key Laboratory of Mechanical System and Vibration, School of Mechanical Engineering, Shanghai Jiao Tong University, 200240 Shanghai, China. ✉e-mail: zhimiaoy@sjtu.edu.cn

Here, we present a reprogrammable mechanical metamaterial consisting of hetero-element functional groups with complete–elastic and rigid–elastic mechanical behaviors. Our study demonstrates continuous stiffness modulation and continuous regulation of Poisson's ratio with multimode deformation by functional group transformation and ring reconfiguration. The demonstrated properties of this reprogrammable mechanical metamaterial hold potential implications in intelligent systems such as reprogrammable mechanical computing, multipurpose robots, and advanced morphable vehicles.

## Results

### Design concept

The mechanical metamaterials composed of periodically arranged homogeneous elements, such as complete–elastic (C) elements with positive stiffness and rigid–elastic (R) elements with negative stiffness, are not mutually transformable (Fig. 1a). This leads to limited reprogramming functions. To enable rich reprogrammable mechanical performances, heterogeneous C and R elements are ingeniously coupled to constitute a C or R functional group. The same type of functional groups is utilized to create the C or R-ring metamaterials. The C and R elements can transfer between adjacent functional groups of a ring, allowing for reversible mutual transformation between the C and R-ring metamaterials. The ring metamaterial composed of heterogeneous elements can serve as the cell unit to constitute periodic homogeneous metamaterials.

The size of the periodic unit structure, i.e., the ring metamaterial, can be minimized to three. The geometry of the triangular ring metamaterial exhibits triangular stability parameters of $\Omega = 0°$ and $\Gamma = 60°$ for the C-ring metamaterial or $\Omega = 60°$ and $\Gamma = 0°$ for the R-ring metamaterial. Meanwhile, for the quadrilateral ring metamaterial, its geometry can be adjusted based on the parameter angle $\Gamma$ for the C-ring metamaterial or $\Omega$ for the R-ring metamaterial (where $0° \leq \Omega/\Gamma \leq 180°$) due to its parallelogram structural features, offering more reconfiguration and greater re-programmability than the triangular ring.

The quadrilateral ring metamaterial with both rotational and axial symmetries was designed to exhibit torsional deformation, axial deformation with zero Poisson's ratio and auxetic deformation with negative Poisson's ratio (Fig. 1b). The ring's multimode deformation is facilitated by the compressible/tensible C elements and compressible R elements, allowing it to undergo contraction, expansion, or both. The geometry of the quadrilateral ring is able to transform topologically from square to rhombus and parallel lines, referred as the ring reconfiguration. Taking the R-ring metamaterial for instance, its parameter angle $\Omega$ is tuned within the ring reconfiguration range of [0°, 180°] (Fig. 1c). Through the ring reconfigurations, nonlinear stress–strain curves of both the C and R metamaterials, respectively having positive and negative stiffness, can be continuously reprogrammed.

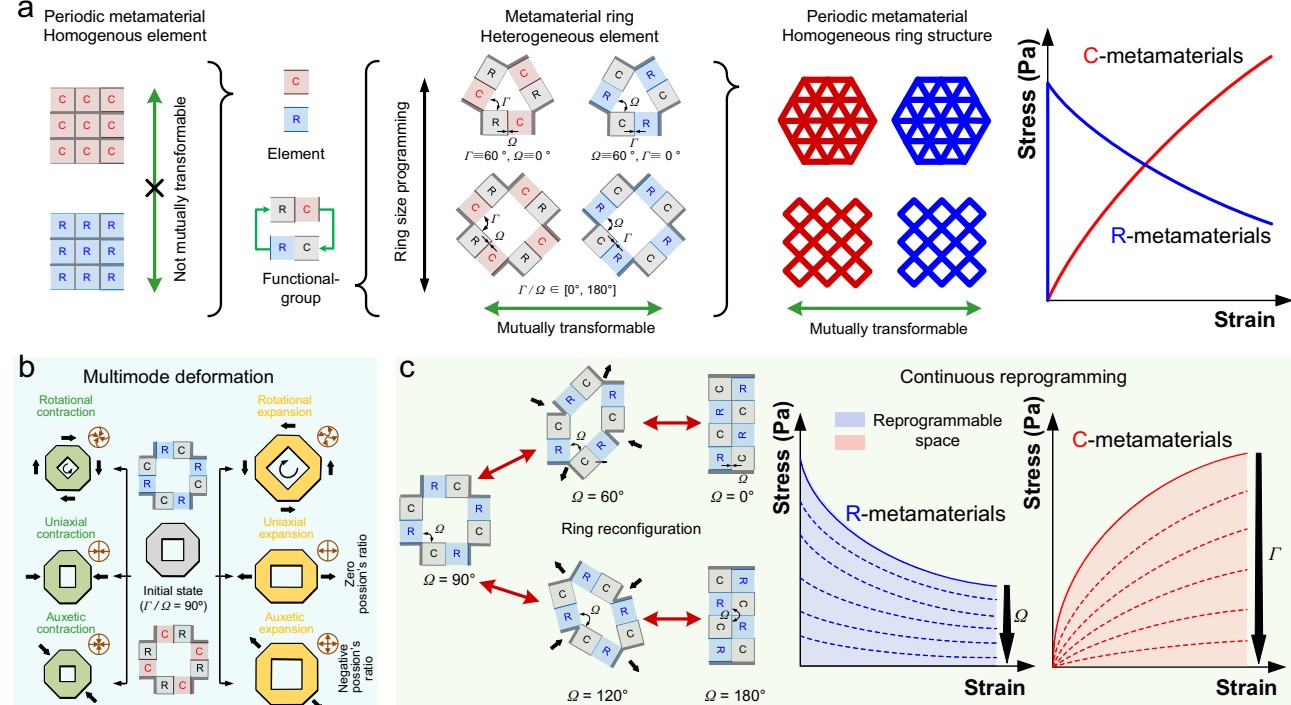

**Fig. 1 | Design concept of reprogrammable mechanical metamaterials with functional group transformation and ring reconfiguration. a** Comparison between mechanical metamaterials composed of homogeneous elements and heterogeneous elements: mutually non-transformable behavior is observed within single-element (either red-colored complete–elastic C or blue-colored rigid–elastic R) periodic homogeneous metamaterials. Conversely, the C or R functional group composed of one C and one R elements can mutually transform. Deformation occurs exclusively along the normal direction of the colored solid element sidelines. Design and construction of triangular and quadrilateral ring metamaterial employ alternating heterogeneous C and R elements. Symbols $\Omega$ and $\Gamma$, respectively, represent the angular relationship between adjacent elements within ring metamaterials. Ring metamaterials (C ring and R ring) composed of corresponding functional groups can interconvert. Periodic homogeneous metamaterials composed of C or R-ring metamaterials can also interconvert, thereby establishing interchangeability of stress–strain relationships between C and R metamaterials. **b** The schematic of multimode deformation of designed mechanical metamaterials with an example of the quadrilateral ring (light cyan area). Contraction and expansion deformations are shown in dark green and dark yellow blocks, respectively. The rotational, uniaxial (zero Poisson's ratio), auxetic (negative Poisson's ratio) contraction, and expansion shown in the same row have the same force action mode but opposite direction (black arrows). **c** The schematic of quadrilateral ring reconfiguration (example: R-ring metamaterial with $\Omega = 0°$, 60°, 90°, 120°, and 180°) and the continuously reprogrammable mechanical responses for C and R metamaterials by ring reconfigurations (light green area). The light blue and red shadings represent the reprogrammable spaces for R and C metamaterials, respectively.

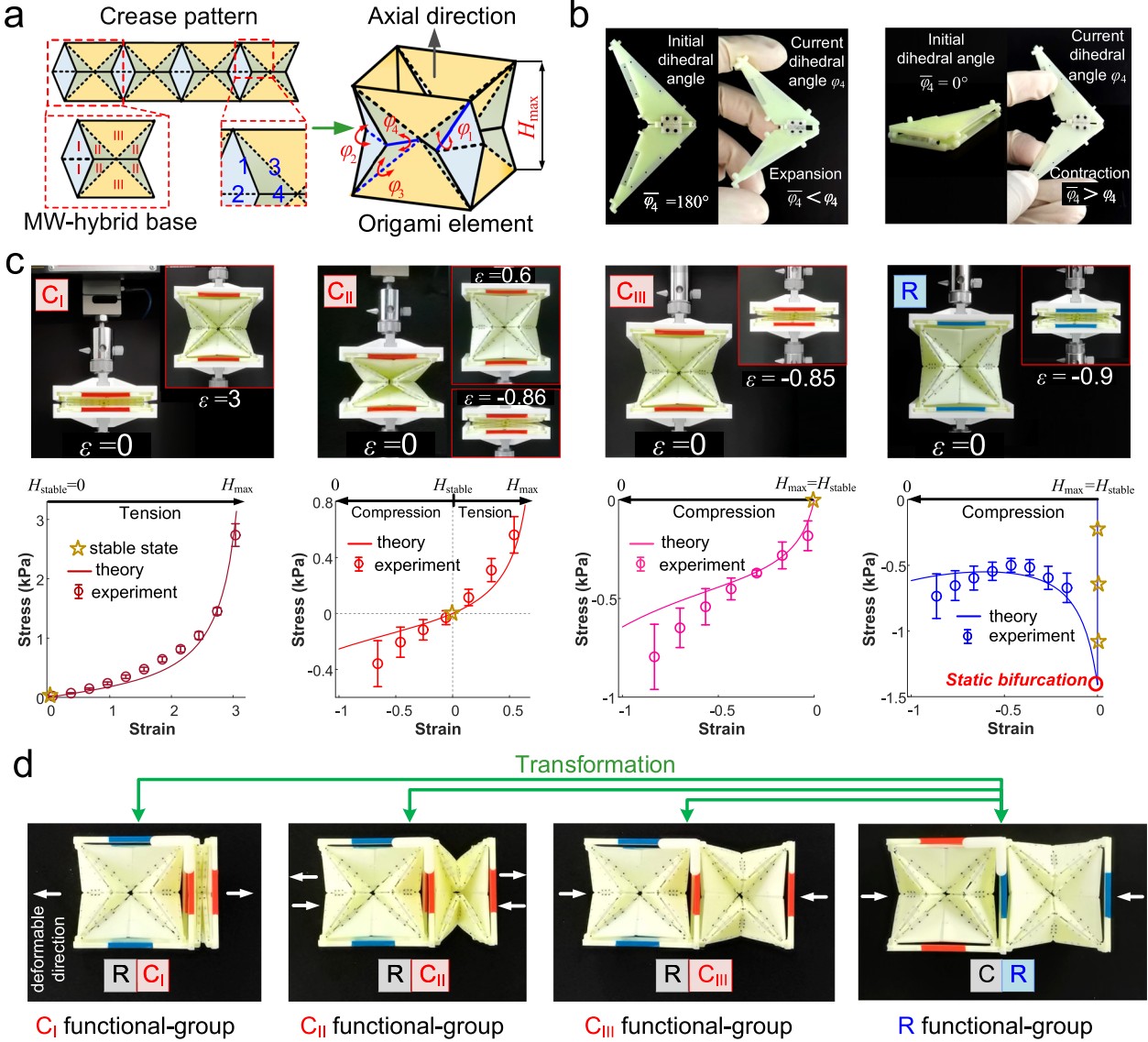

**Fig. 2 | The designed origami element and constructed functional group. a** The crease pattern and 3D schematic of designed origami element. The base with three classes of facets is displayed in the left red dashed box. Each kind of creases are displayed in the right red dashed box. The black arrow points the axial direction of the element. Symbols $\varphi_i$ represent the independent angle parameter of the origami element, and $i$ corresponds to the crease 1, 2, 3, and 4. $H_{max}$ represents the maximal height of the designed origami element. **b** The physical demonstration of expansion and contraction tendency of crease segments under different initial dihedrals (example: crease 4). The facets with high stiffness are manufactured using photosensitive resin by light curing three-dimensional (3D) printing technique for mechanical testing. Creases with representative angles are produced through fused deposition modeling 3D printing technique of thermoplastic polyurethane (TPU) with a hardness of 95 A. See Supplementary Movie 9 and "Methods" for the details about origami element fabrication and assembly. **c** The snapshots and mechanical responses of the $C_I$, $C_{II}$, $C_{III}$, and R elements. The deformable modes of origami elements from left to right are full-length tension ($C_I$), partial-length tension and compression ($C_{II}$), full-length compression ($C_{III}$), and full-length compression with a static bifurcation behavior (R). The snapshots display the physical prototypes of origami element in stable state and maximum deformation state. The selected initial dihedral angle for creases of the physical prototype is listed in Supplementary Fig. 13b and the side length $l$ of the physical prototype is 110 mm. The error bars present the upper and lower limits of stress for multiple experimental subjects testing. $H_{stable}$ represents the structural height of designed origami element at stable state. Source data are provided as a Source Data file. **d** The prototype demonstration and transformation schematic of $C_I$, $C_{II}$, $C_{III}$, and R functional group constructed by one C ($C_I$, $C_{II}$, or $C_{III}$) element and one R element shown in (**c**). The white arrow indicates the deformable direction of the functional group.

## Origami heterogeneous elements

The complete−elastic and rigid−elastic elements are created by origami mechanics. The crease pattern of the origami element is designed as the herringbone arrangement of the Miura-Waterbomb (MW) hybrid base, which is a fusion of Miura-ori and waterbomb bases (Supplementary Discussion 1), depicted by four creases 1, 2, 3, 4, and three facets I, II, III (Fig. 2a). Via the head-tail connection of four MW-hybrid bases, a centrosymmetric tubular origami element is formulated. Such an origami element is classified to the volumetric origami with approximately synchronous rigid folding and pseudo-single

degree of freedom folding/unfolding at the axial direction[19]. Consequently, the kinematics of the origami structure can be analyzed at height $H$ by independently performing rigid folding of Miura-ori and waterbomb bases, with $\varphi_1$, $\varphi_2$, $\varphi_3$, and $\varphi_4$ describing the current dihedral angles between adjacent facets of the four crease segments (Supplementary Discussions 2 and 3).

The mechanical response of the origami element is dependent on changes in dihedral angles from their initial morphologies ($\bar{\varphi}_1, \bar{\varphi}_2, \bar{\varphi}_3, \bar{\varphi}_4$) to their current morphologies ($\varphi_1, \varphi_2, \varphi_3, \varphi_4$) (Supplementary Discussion 5). The initial dihedral angle is in a state without internal

stress and able to be varied between 0° and 360°. While folding or unfolding, the current dihedral angle of the mountain crease changes within (180°, 360°], and the current dihedral angle of the valley crease alters within [0°, 180°). If the current dihedral angle differs from the initial dihedral angle, the crease is in a constrained state and exhibits a tendency to recover from its current morphology to its initial morphology. For mountain creases, when $\varphi_1 < \bar{\varphi}_1$ or $\varphi_4 < \bar{\varphi}_4$, the crease segment has a contraction effect on the overall origami structure. Conversely, when $\varphi_1 > \bar{\varphi}_1$ or $\varphi_4 > \bar{\varphi}_4$, the crease segment leads to an expansion effect on the origami. For valley creases, a contraction effect and an expansion effect on the origami are generated when $\varphi_2 > \bar{\varphi}_2$ or $\varphi_3 > \bar{\varphi}_3$ and $\varphi_2 < \bar{\varphi}_2$ or $\varphi_3 < \bar{\varphi}_3$, respectively. By assembling 3D-printed facets and creases, the expansion and contraction tendencies of the dihedral angle are realized (Fig. 2b and Supplementary Movie 1).

Given that the structural axial deformation-induced strain primarily takes place at the crease, the facet deformation is not taken into account. In other words, the crease is treated as a compliant flexural pivot. The Hookean energy expression[20] is utilized to calculate the potential energy $U$ of the origami structure, along with the reaction force that acts in the axial direction (Supplementary Discussion 4). Integrating the different mechanical contributions of the four creases with the contraction or expansion potential energy, the origami element exhibits complete–elastic (C) mechanical response with $U_{\mathrm{exp}} = U_{\mathrm{con}}$ or rigid–elastic (R) mechanical response with $U_{\mathrm{exp}} > U_{\mathrm{con}}$ (Supplementary Discussion 6).

The axial contraction tendency of the C element overwhelms its axial expansion tendency when its current height $H$ is greater than the stable height $H_{\mathrm{stable}}$. To maintain the current height, the structure is in tension. Conversely, the C element is in compression when the current height $H$ is smaller than the stable height $H_{\mathrm{stable}}$. Therefore, based on deformable types, the C element can be subdivided into three categories: $C_I$, which is tensible, with $H_{\mathrm{stable}} < H \le H_{\max}$; $C_{II}$, which is compressible or tensible, with $0 \le H < H_{\mathrm{stable}}$ or $H_{\mathrm{stable}} < H \le H_{\max}$; and $C_{III}$, which is compressible, with $0 \le H < H_{\mathrm{stable}}$ (Fig. 2c and Supplementary Movie 2).

The R element is full-length compressible as $C_{III}$ but with a static bifurcation occurring at $H_{\max}$ (Fig. 2c). Prior to bifurcation, the geometric restriction in the origami girdling form results in the facets being in tension to counterbalance the extra expansion effect caused by the bending stress of creases. Facet participation in achieving structural equilibrium contributes to the stress-stiffening of the creases in the current morphology and enhances the axial bearing capacity of the R element. The maximal axial bearing capacity, which determines the force threshold of deformation, depends on the extra expansion effect of the creases when the tensile force on the facets is reduced to zero (Supplementary Discussion 7). Following bifurcation, the R element undergoes unidirectional deformation via inward compression without load sharing by the facets. In other words, the R element exhibits a dual-mode behavior that combines the characteristics of both rigid and elastic mechanical systems: load-bearing before bifurcation and deformation after bifurcation[21].

## Functional group transformation

Although the axial deformation modes of the C and R origami elements differ, both types of elements can be flat-folded. The transverse deformation caused by elastic bending of the facets without origami folding is negligible compared to axial deformation (Supplementary Fig. 25). The C and R origami elements demonstrate similar anisotropic deformation characteristics. To create a functional group, a complete–elastic ($C_I$, $C_{II}$, or $C_{III}$) origami element is connected to a rigid–elastic (R) origami element (Fig. 2d), in a way that the anisotropic folding of the axial and transverse directions of the origami elements align, with an elastic hinge connecting one end of the coincidence line (Supplementary Fig. 27d). Consequently, the axial deformation of the C functional group is the folding/unfolding of the C element, while the

axial deformation of the R functional group is the folding of the R element.

With the elastic hinge, it is possible for one origami element to rigidly rotate around another, allowing for a free mutual transformation between the C and R functional groups (Supplementary Movie 3). In the C functional group, the complete–elastic ($C_I$, $C_{II}$, or $C_{III}$) origami element remains its stable height (Fig. 2d) for the construction of a C functional polygonal ring metamaterial. As C functional group transforms to R functional group, the C element participates in the recombination of the functional group with its maximum height. As a result, the C element maintains the same dimensions as the R element in the R functional group transformed from the C functional group, enabling the construction of a R functional polygonal ring metamaterial. The hinges that allow rotation, both within and between the functional groups of a ring, are designed to be parallel to each other. This allows for better coordination of movement during the transformation of the functional group. Additionally, the two hinges closest to each other are positioned diagonally on the closed side of the origami element. This arrangement helps to prevent any physical interference that may occur during the transformation of the functional group.

A triangular C-ring metamaterial can be formed by connecting three identical $C_I$, $C_{II}$, or $C_{III}$ functional groups with rotatable hinges (Fig. 3a). In the C-ring metamaterial, we define the angle of the inter-group hinges as $\Gamma$ and the angle of the intra-group hinges as $\Omega$. When a torque is applied in the $\mathbf{z}$ axis to the C-ring metamaterial ($\Gamma = 60°$, $\Omega = 0°$), it undergoes either torsional expansion ($C_I$), torsional contraction ($C_{III}$), or both ($C_{II}$), depending on the complete–elastic types of the origami element (Supplementary Movie 4). By exchanging the angles between intra-group and inter-group elements ($\Omega$ ranging from 0° to 60° and $\Gamma$ ranging from 60° to 0°), a triangular C-ring metamaterial can be transformed into a triangular R-ring metamaterial (Supplementary Movie 5), and vice versa. The triangular R-ring metamaterial deforms through torsional contraction.

The triangular ring metamaterial possesses rotational symmetry, resulting in regular hexagonal topological outer contour with a regular triangular inner cavity (Fig. 3b). The theoretical relationships between the normalized sweeping angle ($\Delta\Theta/\Delta\Theta_{\max}$) and the radial stretch of the cavity ($R_i/R_{i0}$) or the outer contour ($R_o/R_{o0}$) are demonstrated with experimental data in Fig. 3b and Supplementary Fig. 31. As the sweeping angle $\Delta\Theta$ increases, the stretch $R_i/R_{i0}$ and $R_o/R_{o0}$ exhibit a weakly nonlinear increase for torsional expansion or a weakly nonlinear decrease for torsional contraction. It is important to note that, a reconstruction of functional groups needs to be performed at the position of $\Delta\Theta/\Delta\Theta_{\max} = 0$ in order for deformation curve transitions between $C_I$ and R rings, $C_{II}$ and R rings, $C_{III}$ and R rings. Consequently, the point of $\Delta\Theta/\Delta\Theta_{\max} = 0$ is discontinuous for deformation involving functional group transformation while it is continuous for deformation of the same type of ring metamaterial.

The torque of the triangular ring metamaterial exhibits a similar variation trend with the radial strain as seen in the stress–strain relationship of the origami element (Fig. 3c). Different deformable origami elements within ring metamaterials result in diverse mechanical properties. For instance, all C-ring metamaterials maintain positive stiffness deformation, while the R-ring metamaterial exhibits negative stiffness after a torque threshold of −2.02 N·mm. This demonstrates that the mechanical characteristics of the ring metamaterials can be reprogrammed in situ through functional group transformation.

By increasing the ring size to four and setting the inter-group angle of 90°, the quadrilateral rotationally symmetrical ring metamaterials are constructed (Supplementary Fig. 30). Like triangular ring metamaterials, they are able to transform between the C ring and R ring (Supplementary Movie 6), performing overall torsional expansion and contraction under the torque around the $\mathbf{z}$ axis. The torque variation of quadrilateral ring metamaterials is 1.4 times that with triangular ring metamaterials (Supplementary Fig. 31d). We assessed the

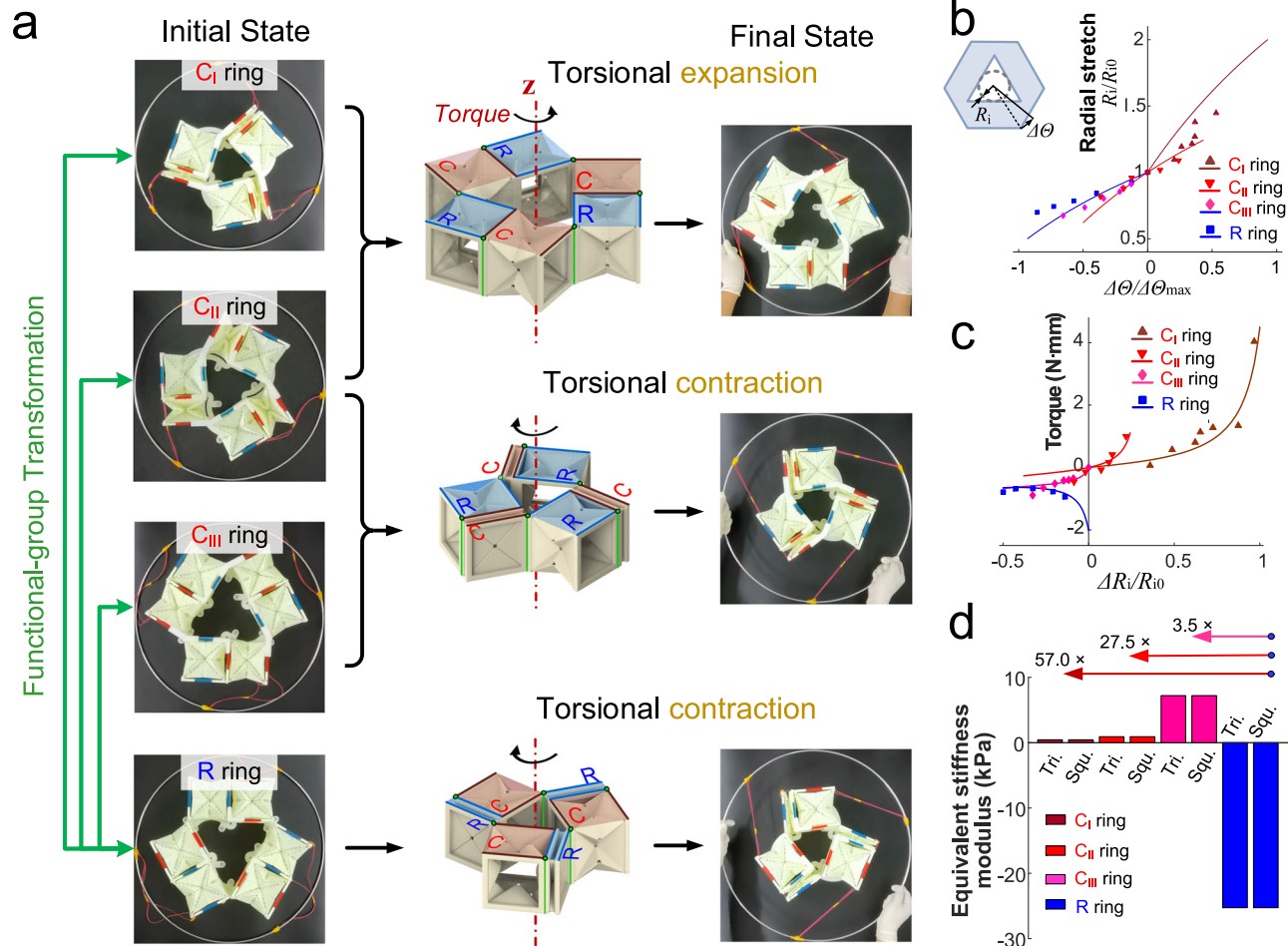

**Fig. 3 | The torsional deformation and mechanical characterization of triangular ring metamaterials. a** The prototype snapshots of initial stable state and the final deformed state of designed triangular ring metamaterials. The ring metamaterials in initial state from top to bottom are constructed by $C_I$, $C_{II}$, $C_{III}$, and R functional groups. The geometry of triangular ring metamaterials is fixed by 3D-printed plastic structural supports, see detail in Supplementary Fig. 27. The outer iron ring and the cords arranged at the evenly divided position are used to apply the external force at outer wall of ring metamaterials. The rendered images in the middle column show the force applied form during deformation. The green arrow indicates functional groups that can be transformed to each other. **b** Radial stretch $R_i/R_{i0}$ varied as a function of sweeping angle ratio $\Delta\Theta/\Delta\Theta_{max}$. $R_i$ and $\Delta\Theta$ are, respectively, the inner tangent circle radius of the inner cavity and sweep angle under torsional deformation. The solid lines represent theoretical results, and the discrete points denote experimental results. Source data are provided as a Source Data file. **c** Semi-experimental torque–radial strain results of triangular ring metamaterials. The abscissa $\Delta R_i/R_{i0}$ represents the ratio of the variation of the inner radius, as characteristic size, of the metamaterial to its initial inner radius and denotes the difference between deformed radial stretch and initial radial stretch. The negative value corresponds to the torsional contraction deformation, and the positive value corresponds to the torsional expansion deformation. **d** Comparison of equivalent stiffness modulus between triangular rings and those of quadrilateral rings. The equivalent stiffness modulus is calculated by Supplementary Eq. (44).

equivalent stiffness modulus of triangular and quadrilateral rings by examining the ratio of circumferential stress to radial strain at initial deformation (Fig. 3d). The equivalent stiffness modulus of quadrilateral rings is identical to that of triangular rings regardless of the functional groups. After the functional group transformation, the sign of the equivalent stiffness modulus is reversed, and the absolute value of R-ring modulus is 57.0, 27.5, and 3.5 times that of $C_I$ ring, $C_{II}$ ring, and $C_{III}$ ring, respectively.

In addition to torsional deformation, the quadrilateral ring exhibits axial deformation with a Poisson's ratio of 0 and auxetic behavior with negative Poisson's ratio (Fig. 4a and Supplementary Figs. 32 and 33). The quadrilateral $C_I$ or $C_{II}$ ring elongates with zero Poisson's ratio when stretched in the **x** or **y** direction and expands bilaterally upon being uniaxially stretched in the direction of **x** + 45° or **y** + 45° with a negative Poisson's ratio of −1 (Supplementary Movie 7). On the other hand, the quadrilateral $C_{II}$ or $C_{III}$ ring contracts under compressive force in the **x** + 45° or **y** + 45° direction with a negative Poisson's ratio of −1 and shortens with zero Poisson's ratio under compression in the **x** or **y** direction.

The quadrilateral C ring can be transformed to the quadrilateral R ring (Fig. 4a). The quadrilateral R ring exhibits biaxial contraction with a Poisson's ratio of −1 and uniaxial shortening with a Poisson's ratio of 0. The experimental results of the stress–strain curves for the quadrilateral ring under these deformations align with the theoretical analysis for all tested specimens (Fig. 4b). In Fig. 4c, the elastic modulus at initial strain of auxetic behavior (Poisson's ratio of −1) for $C_I$, $C_{II}$, $C_{III}$, and R rings are 0.5 times those of uniaxial deformation (Poisson's ratio of 0). After functional group transformation, the elastic modulus of R ring changes sign and its magnitude is 28.5, 22.1, and 3.5 times that of $C_I$, $C_{II}$, and $C_{III}$ rings, respectively.

## Ring metamaterial reconfiguration

The reconfiguration of the quadrilateral C rings can be achieved by adjusting the range of $\Gamma$ values in (0°, 90°) ∩ (90°, 180°) with centrosymmetry but not rotationally symmetry. Due to the difficulty of transmitting circumferential shear force along the splicing direction in each functional group, the quadrilateral C rings with $\Gamma \in (0°, 90°) \cap (90°, 180°)$ cannot undergo torsional deformation. One of the

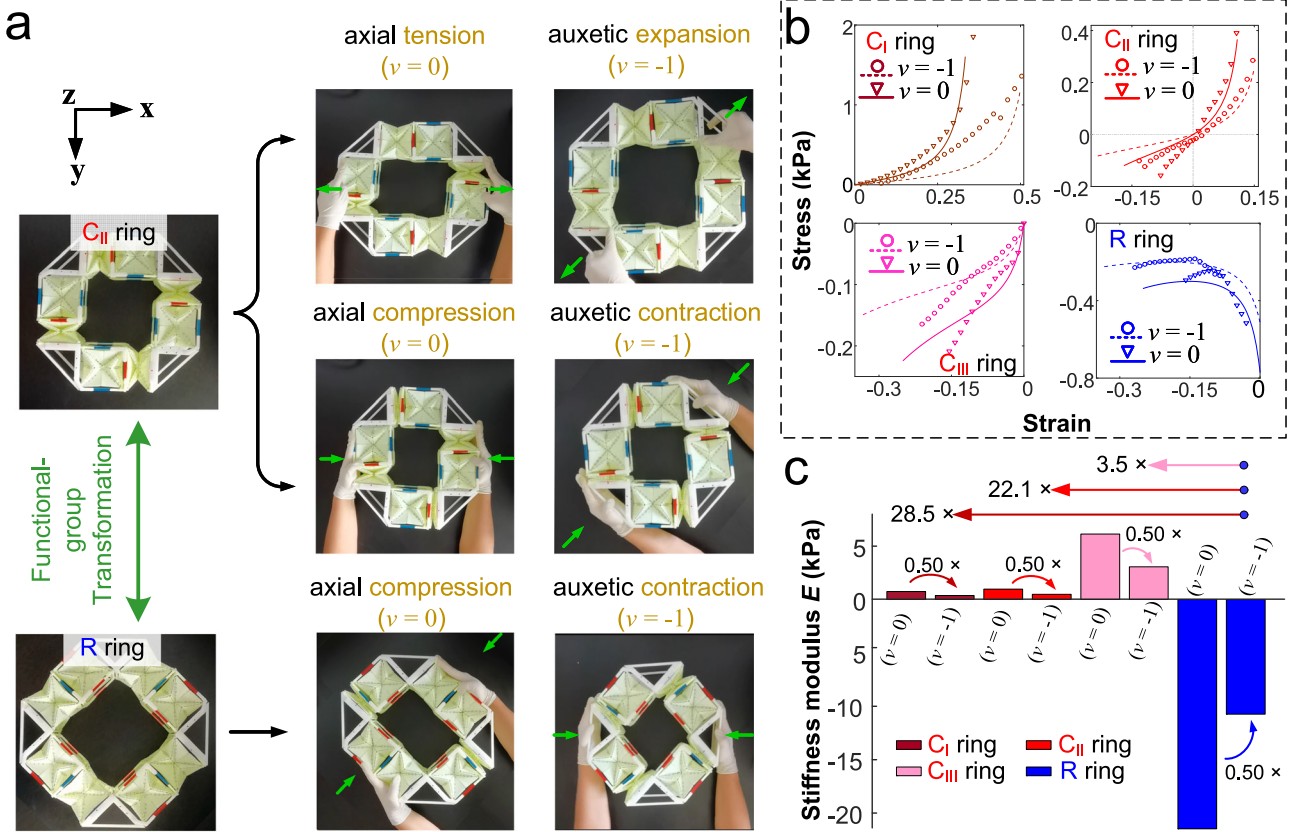

**Fig. 4 | The multidirectional axial deformation and mechanical characterization of quadrilateral ring metamaterials. a** The prototype snapshots of designed quadrilateral rings with $C_{II}$ and R functional groups at initial stable state, the final states with axial deformation with Poisson's ratio $v = 0$, and auxetic deformation with $v = -1$. The geometry of quadrilateral ring metamaterials is fixed by 3D-printed plastic structural supports, see detail in Supplementary Fig. 27. The green arrows in snapshots point out the forced direction. **b** The stress–strain curves and **c** Comparison of stiffness modulus of the quadrilateral ring constructed by $C_I$, $C_{II}$, $C_{III}$, and R functional groups under multidirectional axial deformation. The solid lines and hollow dots in (**b**) represent theoretical results and experimental results, respectively. Source data in (**b**) are provided as a Source Data file.

evolutionary configurations, named morphology I, is depicted in Fig. 5a, Supplementary Figs. 34 and 35 (Supplementary Movie 6). The auxetic characteristic with a negative Poisson's ratio along the symmetrical axis is maintained (Supplementary Movie 8). By adjusting the tuning parameter $\Omega$ within the range of $(0°, 90°) \cap (90°, 180°)$, the quadrilateral R ring can reconfigure itself into the evolutionary morphology I. When subjected to compressive force along the symmetrical axis, it exhibits a negative Poisson's ratio.

The theoretical results of the force-displacement relationship are compared with the experimental data shown in Fig. 5b for R ring and Supplementary Fig. 36 for $C_I$, $C_{II}$, and $C_{III}$ rings. As the angle $\Gamma$ or $\Omega$ increases in the range of $(0°, 180°)$, the reaction force of the quadrilateral ring in evolutionary morphology I gradually decreases to nearly 0 N, indicating its continuously reprogrammable mechanical properties. The initial stiffness modulus of quadrilateral ring decreases as the angle $\Gamma$ or $\Omega$ increases (Fig. 5c), with ranges of $(0.377, 0.754)$ kPa, $(0.709, 1.587)$ kPa, $(1.533, 6.129)$ kPa and $(−21.466, −5.372)$ kPa for $C_I$ ring, $C_{II}$ ring, $C_{III}$ ring and R ring, respectively. Through ring reconfiguration, the stiffness modulus can be smoothly adjusted by four times for R-ring metamaterial. Furthermore, with the participation of functional group transformation, the absolute value of $E_{max}/E_{min}$ can reach up to 57.

By the further ring reconfiguration ($\Gamma = 0°$ or $180°$), the quadrilateral C ring shapes to evolutionary morphology II (Fig. 5d). In morphology II, the quadrilateral C ring undergoes uniaxial deformation, with elongation in $C_I$ or $C_{II}$ under tension, and shortening in $C_{II}$ or $C_{III}$ under compression. The Poisson's ratio is zero. Similarly, when the

quadrilateral R ring shapes to morphology II by the ring reconfiguration ($\Omega = 0°$ or $180°$), it shortens with a Poisson's ratio of zero under uniaxial compression. The stress–strain curves for tension deformation keep monotonous (Fig. 5e). On the contrary, the stress–strain curves for compression deformation exhibit two extreme points, particularly for R-ring metamaterial, which reflects the compression characteristics of alternating folding of the deformable origami elements within the metamaterial.

**Periodic homogeneous metamaterial**

The negative Poisson's ratio (auxetic) effect primarily occurs in stretch-dominated structures where the shear stiffness is greater than the axial stiffness[22–25], rigid units with relative rotation[26–28], chiral lattices with node rotation and ligament bending[24,29], and sheet-like origami with reoriented structural elements[28,30]. The R/C functional group constructed in this study belongs to a stretch-dominated structure and achieves the microstructure concept of a shock absorber element with a negative Poisson's ratio, as described in literature[22]. In constructing two-dimensional material networks, this functional group, simplified as a spring, can be more easily arranged in various polygon network structures, such as triangles or quadrilaterals, leading to the fabrication of more diverse material structures and properties, compared to re-entrant structures[5], rotating cube structures[28,31–33], and Miura origami[6].

Although the triangular ring metamaterial is difficult to deform under uniaxial force, the metamaterial periodically arranged with triangular ring with a form of planar network is easy to deform under

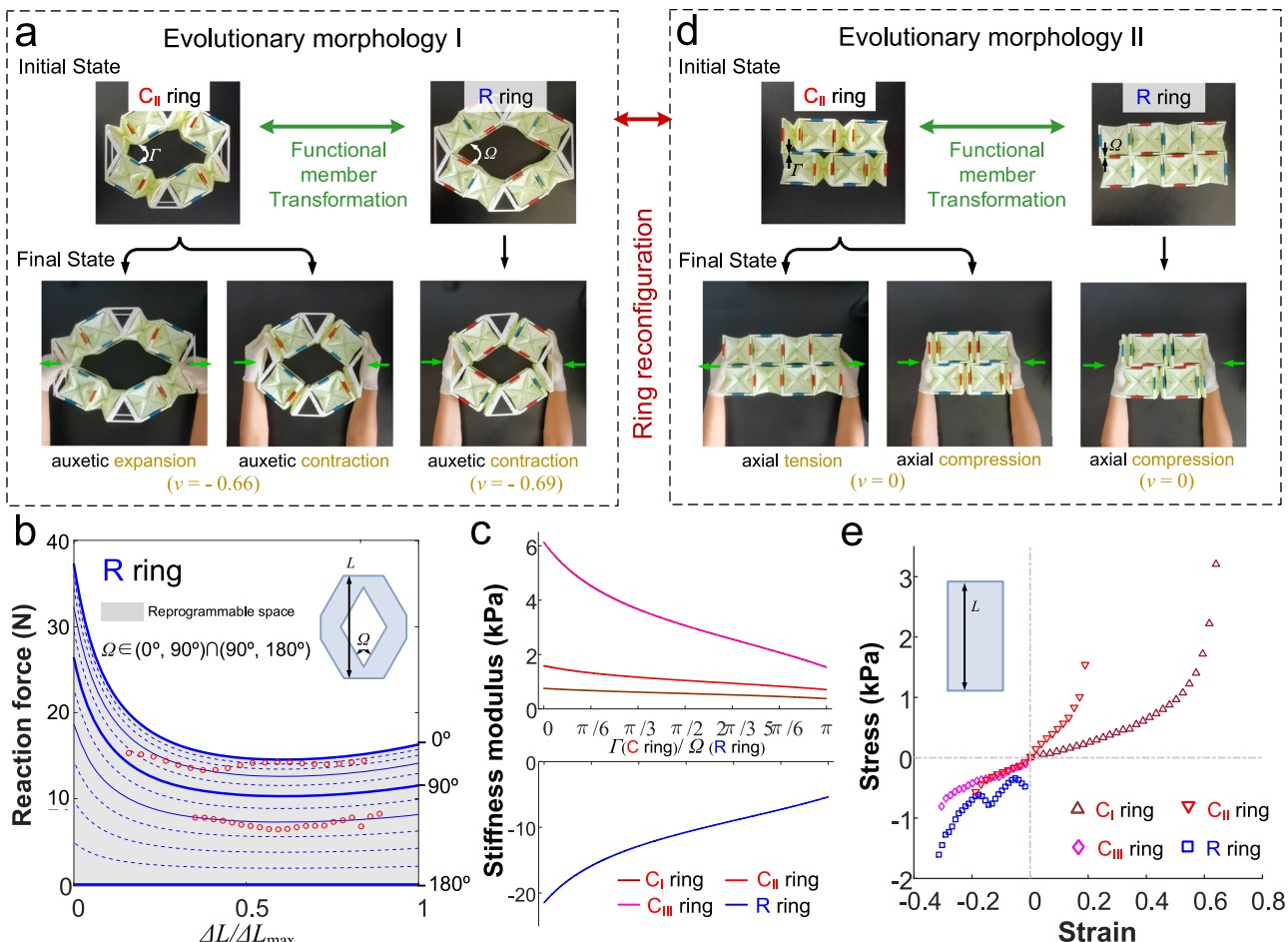

**Fig. 5 | Two evolutionary morphology of quadrilateral ring metamaterials and their mechanical properties. a** The prototype snapshots of initial stable state and auxetic deformation of designed quadrilateral rings with complete−elastic (example: $C_{II}$ ring with $\Gamma$=60°) functional groups in evolutionary morphology I. And those of transformed ring metamaterials with rigid−elastic (R) functional groups in evolutionary morphology I. **b** The continuous reprogrammable force-displacement curve of quadrilateral ring metamaterial with rigid−elastic functional groups as the variation of adjustable angle $\Omega$. The solid lines represent the theoretical results, while the hollow points represent the experimental results. **c** The continuous

reprograming of stiffness modulus at the initial state for reprogrammable mechanical metamaterial with quadrilateral ring in evolutionary morphology I. **d** The prototype snapshots of initial stable state and axial deformation with Poisson's ratio $v = 0$ of designed quadrilateral rings with complete−elastic (example: $C_{II}$ ring with $\Gamma = 0°$) in evolutionary morphology II. And those of transformed ring metamaterials with rigid−elastic (R) functional groups in evolutionary morphology II. **e** The experimental stress−strain curves of quadrilateral rings constructed by $C_I$, $C_{II}$, $C_{III}$, and R functional groups in evolutionary morphology II. Source data are provided as a Source Data file.

uniaxial tension and compression. The triangular ring periodic homogeneous metamaterial is considered as a material consisting of randomly packed smooth spherical particles[22]. Its Poisson's ratio is a constant of −1, related to the stiffness ratio of non-deformable to deformable directions of functional groups[34–36].

The Poisson's ratio of periodic homogeneous metamaterial of quadrilateral ring is determined by the constitutive relationship when considering the functional group as the representative volume element[37–39]. See detail for constitutive relationship in Supplementary Discussions 8C. Since the component of compliance tensor $S_{ij}$ is greater than zero for adjustable angle within (0°, 90°), the periodic homogeneous metamaterials of quadrilateral ring always exhibit auxeticity for this range of adjustable angle[37]. The Poisson's ratio can be reprogrammed with the element's stable-state height $H_{stable}$ and inter-group angle $\Gamma$ or $\Omega$ of the ring configuration. Analytical contour plots show that the Poisson's ratio decreases from 0 to −∞ as the inter-group angle increases and the origami stable height decreases (Supplementary Fig. 37).

Two prototypes of periodic homogeneous metamaterials composed of quadrilateral C rings were fabricated, see "Methods" for details. Instead of replacing fixed supports for the ring metamaterial,

3D-printed holes were added at the angle adjustable hinges to secure the ring configuration with pins during the reconfiguring process (Fig. 6a and Supplementary Fig. 38). The contraction of the metamaterial with quadrilateral $C_{III}$ ring of $\Gamma = 67.5°$ under uniaxial compression is observed, as shown in the snapshots in Fig. 6b. The $C_{III}$ elements collapse gradually, while the R elements remain unchanged as the strain increases. Inspection of the internal cavity of the ring metamaterial reveals a gradual decrease in transverse size, demonstrating its auxetic characteristics. Similar behaviors are observed for periodic homogeneous metamaterials with larger $\Gamma$ (Supplementary Movie 10). The experimental and theoretical results show a consistent trend of reprogrammable changes (Fig. 6c). The Poisson's ratio of the periodic homogeneous metamaterial with quadrilateral $C_{III}$ ring decreases exponentially from −0.336 to −1.717 with the increase of $\Gamma$ from 22.5° to 157.5°, indicating more pronounced auxetic characteristics for larger $\Gamma$. The experimental results also show slight fluctuations in the Poisson's ratio during compression.

Figure 6d and Supplementary Movie 11 display the expansion of periodic homogeneous metamaterial consisting of quadrilateral $C_I$ rings under axial tension. The synchronous increase in longitudinal and transverse dimensions indicates its auxeticity characteristics.

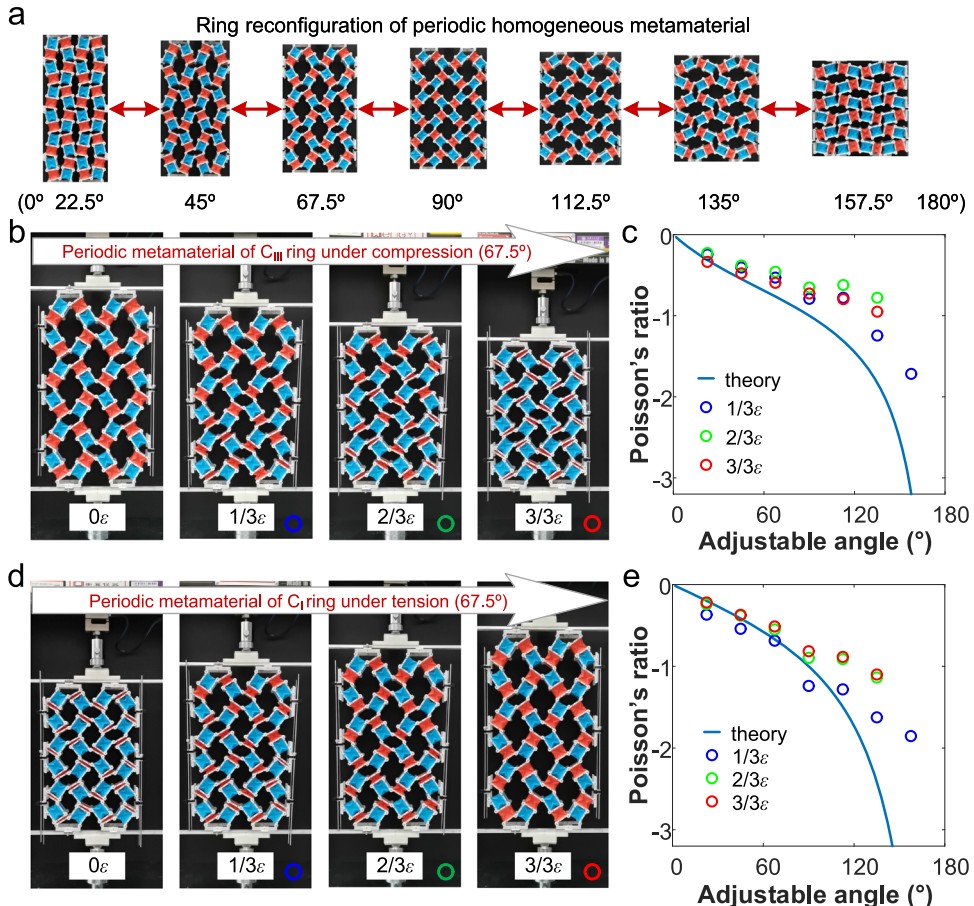

**Fig. 6 | The auxeticity of periodic homogeneous metamaterials. a** The paper prototype of periodic homogeneous metamaterials with quadrilateral ring under different configurations (adjustable angles of 22.5°, 45°, 67.5°, 90°, 112.5°, 135°, and 157.5°). The geometry of quadrilateral ring can be continuous adjusted and fixed by 3D-printed plastic connecting components with detachable pins, see detail in method of Fabrication and experiment testing of periodic homogeneous quadrilateral ring metamaterials. **b** The paper prototype snapshots of an example of periodic homogeneous metamaterial at the strain of 0, $1/3\varepsilon$, $2/3\varepsilon$ and $3/3\varepsilon$ under compression deformation (example: $C_{III}$ rings with the adjustable angle of 67.5°). The side length $l$ of the physical prototype of paper element is 30 mm. See

"Methods" for the details about periodic homogeneous metamaterial fabrication. **c** The Poisson's ratio of periodic homogeneous metamaterial varied as a function of adjustable angle for compression deformation. The solid line represents theoretical results, and the discrete hollow points denote experimental results. **d** The paper prototype snapshots of an example of periodic homogeneous metamaterial at the strain of 0, $1/3\varepsilon$, $2/3\varepsilon$, and $3/3\varepsilon$ under tension deformation (example: $C_I$ rings with the adjustable angle of 67.5°). **e** The Poisson's ratio of periodic homogeneous metamaterial varied as a function of adjustable angle for tension deformation. Source data are provided as a Source Data file.

Similar to the contraction of auxetic behavior, the Poisson's ratio exponentially decreases as $\Gamma$ increases (Fig. 6e). The theoretical and experimental results for the expansion of auxetic behavior align more closely compared to the results for the contraction of auxetic behavior. This is attributed to the more uniform unfolding process of $C_I$ elements in the metamaterial, compared to the folding process of $C_{III}$. By varying the inter-group angle from 22.5° to 157.5°, the Poisson's ratio can be reprogramed from −0.219 to −1.853.

The published tuning of negative Poisson's ratio typically occurs during structural deformation, often accompanied by structural reconfiguration[5,6], making it difficult to maintain a stable configuration under large deformations. In contrast, our experimental validation demonstrates that the negative Poisson's ratio effect persists in stable configurations and can be controlled by adjusting the angles between functional groups, allowing for modifications in the stable configuration and its auxetic behavior. In addition, the rings in the periodic homogeneous metamaterials can independently switch between C and R functional modes without mutual interference. The coexistence of C rings and R rings (see Supplementary Movie 12) can also impact the mechanical performance of the metamaterial. These advantages render this metamaterial suitable for various applications, including load-bearing, soft actuation, and mechanical computing.

## Potential applications

In this section, we present various scenarios illustrating the potential applications of the proposed design paradigm of functional group transformation and ring reconfiguration in intelligent machines and systems, including robots and mechanical computing. For robotics, the designed metamaterial can be utilized to build the body components such as legs and arms (Supplementary Fig. 42a). The unique characteristics of reprogrammable stiffness modulus, achieved through functional group transformation and ring configuration, ensure that the material exhibits low stiffness during motion to provide vibration isolation and high stiffness to support heavy loads[40,41].

This metamaterial also offers different stiffness and strength requirements in orthogonal directions[6]. By utilizing both expansion and contraction deformations, the proposed metamaterials can act as actuators for robots through internal electric drive[42,43] (Supplementary Fig. 42b). The axial and torsional deformations enable the actuator to adapt to different environmental conditions and perform various tasks. For example, the worming caused by axial deformation shown in Fig. 5e enables the actuator to operate effectively in flat, hard contact environments, while rolling caused by torsional deformation in Fig. 3 is suitable for driving equipment in steep or loose contact environment[44,45].

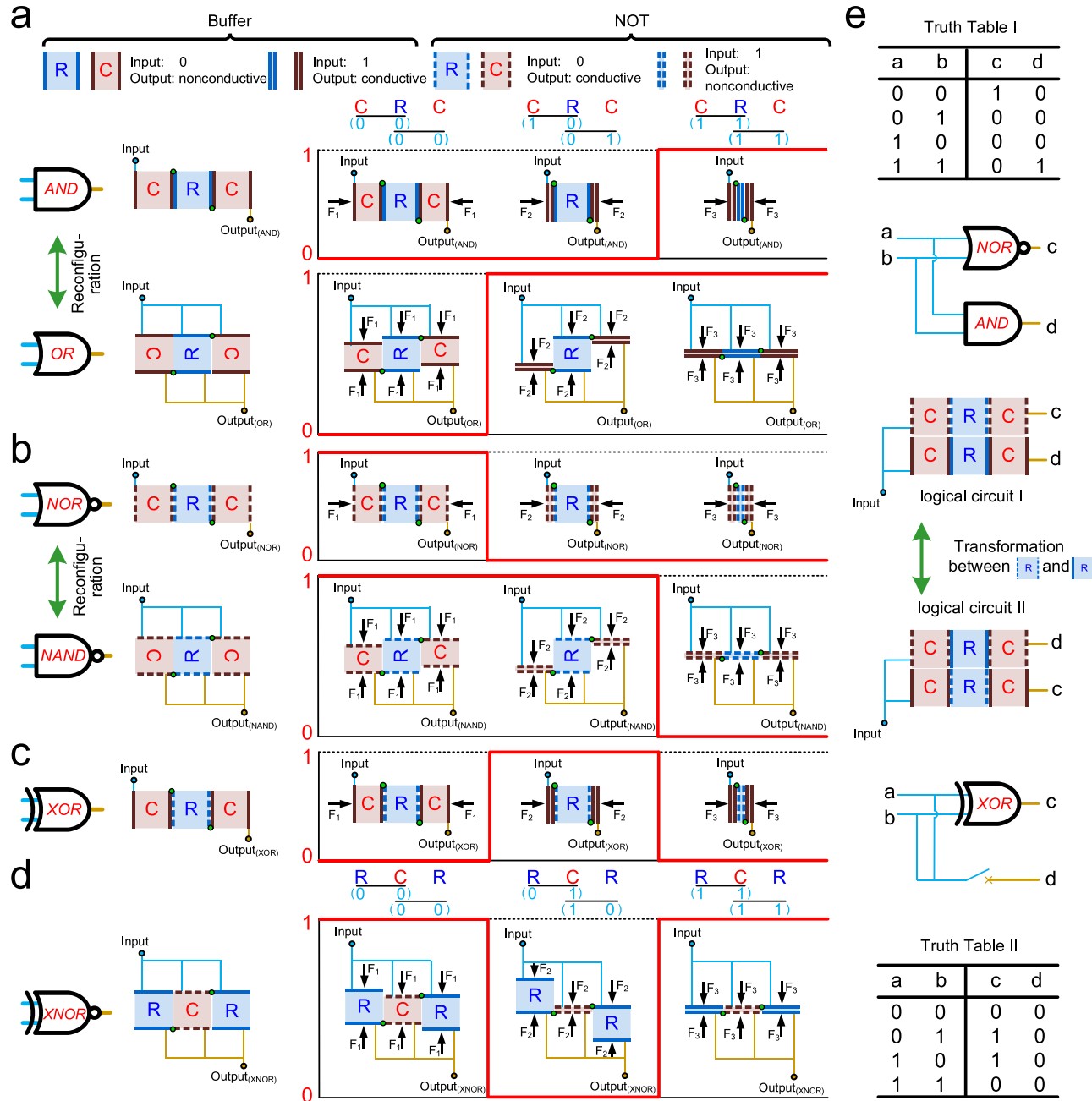

**Fig. 7 | Potential application of reprogrammable mechanical metamaterials in mechanical computing. a** The schematic diagram of Buffer and NOT logic elements, inter-transformable AND and OR logic gates, their input–output responses. **b** The inter-transformable NOR and NAND logic gates, their input–output responses. **c** The XOR logic gates and its input–output responses. **d** The XNOR logic gates and its input–output responses. **e** The schematic diagram of transformable mechanical computing circuits and their Truth Tables.

In addition, the designed metamaterials can be used to anchor machines and facilitate cargo delivery in narrow environments like pipelines, through torsional and axial deformations with negative Poisson's ratio[46] (Supplementary Fig. 42c). The inner cavity of the metamaterial can hold objects, and during the shrinking process, the object becomes anchored to the inner wall of the metamaterial, acting as a grasper. Through functional group transformation, the grasper can adjust its stiffness to safely grip fragile objects with low stiffness or hold hard objects with high stiffness.

By leveraging the discrete mechanical behaviors of origami elements coupled with electrical networks, mechanical computing can be achieved[47,48]. Non-flat-folded mechanical behavior is designated as input "0", while flat-folded mechanical behavior is designated as input

"1" (Fig. 7a). The Buffer origami element exhibits nonconductive output for input "0" and conductive output for input "1", while the NOT origami element exhibits conductive output for input "0" and non-conductive output for input "1", using special electronic contactors (Supplementary Fig. 43). We employ a combination of two adjacent C and R origami elements in different folding states to simulate a 2-bit input. The deformation difference of the C and R origami elements allows the functional group to mimic inputs "0" and "1" when subjected to a pair of forces.

Based on standard Boolean function, the AND logic gate consists of two C Buffers and one R Buffer connected in alternating series, while the OR logic gate consists of two C Buffers and one R Buffer connected in alternating parallel. These functional groups exhibit three folding

modes. The combination of non-flat foldable C and R origami elements represents the input (0, 0), and the combination of flat foldable C and R origami elements represents the input (1, 1). The inputs (0, 1) and (1, 0) are simulated using a combination of flat foldable C and non-flat foldable R origami elements, as they yield the same logical operation result. By arranging hinges at the opposite corners of the R origami element, the logic gate can achieve structural reconfiguration and realize the function conversion between AND and OR gates. Reprogrammable logic devices are promising candidates for future efficient computing systems and adaptive electronics[49,50].

Similarly, NOR and NAND logic gates are constructed using C and R NOT origami elements (Fig. 7b). The NOR and NAND functions also convert under structural reconfiguration. The XOR logic gate is realized with a functional group consisting of two C Buffers and one R NOT connected in alternating series (Fig. 7c). The traditional construction method of XOR logic gate, which requires series-parallel hybrid connection of four or more discrete mechanical behaviors, is simplified by simultaneously inputting (0, 1) and (1, 0) using this approach (Supplementary Fig. 44). This simplifies the complexity of constructing mechanical logic circuits. The simplified XNOR logic gate consists of a functional group with two R Buffers and one C NOT connected in alternating parallel (Fig. 7d).

Figure 7e shows a schematic diagram of a logic circuit in the form of bulk metamaterials and the effect of functional group reconstruction on the logic circuit within the metamaterials. By transforming between R Buffer and R NOT, the logic circuit can switch between AND | NOR logic gate connections and XOR | O circuit connections. The output of the circuit changes dramatically for the same input, as shown in Truth Tables I and II. Acting as a connecting role within a large-scale integrated circuit, this field-reprogrammable gate array also determines whether the logic circuit module runs after the output d, thereby enriching information processing capabilities through functional group transformation[51,52].

## Discussion

We propose a reprogrammable metamaterial design strategy based on functional group transformation and ring reconfiguration with hetero origami elements. Our study demonstrates the reprogramming of complete–elastic (positive stiffness) and rigid–elastic (negative stiffness) performance by functional group transformation. We also show how continuous stiffness modulation and regulation of Poisson's ratio can be achieved by ring reconfiguration. In addition, we obtain multimode expansion and contraction, including torsion, auxetic deformation with negative Poisson's ration, and uniaxial deformation with Poisson's ratio of zero, for different morphologies. These capabilities can be utilized in the construction of robots with multiple task modes and high adaptability to extreme environments, as well as in the design of reprogrammable logic circuits with module function conversion. The reprogramming method of mechanical metamaterial by transformation and reconfiguration design can be applied to various fields, including meta-DNA structures[53], mechanical memories and computing[48], advanced architectures[54], and biomedical devices[55].

## Methods

### Thick facet design of origami element prototype

When the initial dihedral angle of one crease is not equal to its current dihedral angle, it exhibits expansion or contraction effect due to internal stress, leading to the origami element with different stable-state structural heights. If remaining flexible, the facets of the origami element prototype have a large bending deformation under the interaction of the internal stress of all creases. With flexible facets, the prototype will lose the research purpose of the folding characteristics in this work.

The origami element prototype with high stiffness facets is processed by thick facet processing (Fig. 2c). Each facet is treated as an individual and fabricated through the light curing 3D printing technique with high-toughness photosensitive resin (Supplementary Fig. 26). The modulus and strength of the material are 2.7 GPa and 72 MPa, respectively, and the thickness of facets is set to 2 mm to ensure the rigidity. In order to facilitate the assembly between each two facets, the cylinders with circular through hole are arranged at the connection. The radius of cylinder and circular through hole is 1.5 mm and 0.5 mm, respectively. The cylinder on the two connected facets can compose a rotating hinge, replacing crease bending in folding\unfolding, in pairs when two through holes remain concentric. To prevent the interference of the adjacent facets during folding, these cylinders need to be arranged on the side where the dihedral angle of the adjacent facets is less than 180°. In addition, when the height of origami element is gradually decreasing during folding, facet I is always sandwiched by the nearby facet II (Supplementary Fig. 2c). Thus, the connection of facet II needs to be locally thickened. Due to the approximately rigid folding properties of origami element, facet III is split in half. The facet III−1 is provided with a cavity, while facet III−2 is set with partial thin section. When the thin section is inserted into the cavity, only sliding can occur between facet III−1 and facet III−2. The assembled facet III can open or close around the fixed lower endpoint to simulate all facet deformations during folding/unfolding. Square slot is provided on each facet edge of each facet for mounting elastic creases 1, 2, and 3. At the same time, there are screw holes with the diameter of 800 μm at the bottom of the square slot for fixing the crease. For crease 4 on facets II, screw holes are provided directly on rigid facet II to secure the crease material. The elastic creases are processed by fused deposition 3D printing technology using TPU material with the hardness of 95 A. The Young 's modulus of the printed crease material is maintained around 18 MPa. The thickness and width of deformable region of the elastic creases are, respectively, 0.5 mm and 5 mm. However, the creases' length is scaled to 0.36 times those of the thicknessless model.

### Assembly of origami element and their connection in reprogrammable mechanical metamaterials

The materialized origami element is constructed through the head-tail connection of four MW-hybrid bases. The installation positions of the crease are concentrated on both sides of the facets III. When facet III is divided into facet III−1 and facet III−2, the complete component for origami element construction can be assembled by four identical subcomponents. Thus, assembly process of origami element is able to be divided into sub-assembly based on subcomponent and full-assembly of complete component (Supplementary Movie 9). Sub-assembly mainly includes the fixation of all crease materials on facets. To avoid unnecessary physical interference between prototype and tools, we start from the fixation of crease 2, followed by creases 3, then creases 1 and finally creases 4. During sub-assembly, the cylinder pairs at adjacent facet edge are firstly connected by self-tapping screws of M1×3 mm to form rotating hinge and limit the distance between adjacent facet edges (Supplementary Fig. 27). The printed crease is fixed with self-tapping screws of M1 × 3 mm into corresponding square slot one by one. The screw part protruding from the facet shall be cut off with pliers to prevent collision during folding/unfolding. The full-assembly is mainly completed through the internal connection of face III. The thin section of the facet III-2 is inserted into the inner cavity of the facet III-1. The facet III-1 and facet III-2 are connected by connecting strips at their lower endpoints. The M1 × 4 pin is installed through the facet III-1 and facet III-2 to limit the internal opening degree in the facet III.

The shape of an origami prototype with high stiffness facets is approximately a cube. Its length and width are equal, while the height is slightly smaller than the length and width. In order to carry out the connection construction of the metamaterial and realize the sticking different sides together under functional group transformation and ring reconfiguration, the origami element at the maximum height state

must have the same three-dimensional size. Therefore, a hollowed out square plastic frame is added to the top and bottom of the origami element prototype to make up for the height dimension. The plastic frame and the real object are also similarly connected by a rotary hinge. Therefore, a cylinder with a round hole is set at the minimum inner corner of each facet I. A protrusion is also added at each corner of the plastic frame to set cylinder with round hole. The cylinders at the minimum inner corner of each facet I and the protrusion are connected together with screws of M1 × 6 mm. On the opposite narrow sides of the plastic frame, triangular inner grooves are set to install connecting straps to realize the connection between the origami elements in the reprogrammable mechanical metamaterials. The remaining narrow sides are arranged with outwardly convex cylinders to install structural supports and fix the angle of the reprogrammable mechanical metamaterials.

### Uniaxial mechanical tests of origami element and reprogrammable mechanical metamaterials

The mechanical property of each origami element is characterized by uniaxial compression experiments of three prototypes using an electronic universal testing machine (HY-0580, Shanghai Hengyi Testing Machine Co., LTD). During mechanical testing, the prototype of the origami element is vertically mounted in custom-designed clamps, as shown in Fig. 2c. Quasi-static deformation is achieved by applying an upper plate clamp with a single displacement at the rate of 10 mm/min. The reaction force was measured using a 1000 N load cell (BSS-100 kg, Transcell Technology, Inc.) at the sampling frequency of 20 Hz. The load cell and displacement sensor errors are less than 0.03% and 0.5% of the rated output, respectively. Compared with tests of the origami element, the only difference is the connection between test object and fixture for mechanical tests of reprogrammable mechanical metamaterials (Supplementary Fig. 28). The reprogrammable mechanical metamaterials and fixture are connected by pin of M10 × 130 mm or jamming between fixture inwall and convex cylinders of square plastic frame.

### Semi-experimental method of torsional shear stress–strain curve

The torque–radial strain curve is obtained through the semi-experimental method. The semi-experimental method refers to that the torque is calculated based on Supplementary Eq. (43) with experimental data of the origami element and experimental relationship data among height of deformable origami element and inner diameter of reprogrammable mechanical metamaterials. The experimental relationship data among height of deformable origami element and inner diameter of reprogrammable mechanical metamaterials are obtained by image analyzation with Microsoft Visio. The height of deformable origami element and inner diameter of reprogrammable mechanical metamaterials are measured by the size of marked line and circle, respectively (Supplementary Fig. 29). The actual length and radius are equal to the corresponding data obtained from image analysis multiplied by the proportional value. Due to the rotational symmetry of reprogrammable mechanical metamaterials, we analyze the origami element height and the angle variation at each functional group. Then, the mean values are obtained to increase the validity of the data.

### Fabrication and experiment testing of periodic homogeneous quadrilateral ring metamaterials

The reprogrammable auxeticity of quadrilateral ring metamaterials are further investigated with a periodic homogeneous form and paper prototype elements. To ensure the effective use of the experimental space and the better folding of the elements, we set the size of the paper prototype element to 30 mm × 30 mm × 30 mm (length × width × height). And the periodic homogeneous metamaterial is assembled by six-ring metamaterials with an array of three rows and two columns (Supplementary Fig. 38). The paper prototype of element is made by folding red and blue cardboard with pre made creases. The pre made creases on cardboard are cut by cutting plotter. The red paper prototypes represent C elements, and the blue paper prototypes represent R elements. The standard of the cardboard is 180 g/m².

The paper prototype is equipped with two 3D-printed square frames at both ends to fix the shape. There are circular holes with a diameter of 0.8 mm on each side of square frame (Supplementary Fig. 38a). Among them, one set of holes on the opposite sides is used for fixing the square frame and the paper prototype with screws of M1 × 3 mm, and one set of holes on the other opposite sides is used for installing connecting components between elements with screws of M1 × 4 mm. There are two types of connecting components, the first of which is installed on the square frames of C elements. The second ones are installed on that of R elements. The circular plates on one corner of first type of connecting components and the circular plate on one corner of the second type of connecting components overlap and are connected by screws to form a rotating hinge (Supplementary Fig. 38b). Based on this method, the functional group is constructed by one C paper prototype and one R paper prototype. According to the aforementioned functional group connection, the ring metamaterial with functional group transformation is completed (Supplementary Fig. 38c). Circular plates are left on each corner of ring metamaterial to facilitate the connection for constructing periodic homogeneous metamaterial. Sixteen circle holes with a diameter of 0.7 mm and the interval of 22.5° is reserved on lower circular plate of each hinge. Two circle holes with a diameter of 0.7 mm and the interval of 180° is reserved on upper circular plate of each hinge. The circle hole of the upper coincide different circle hole on lower circular plates to adjust the internal angle of the ring metamaterial. These coinciding holes can be fixed by pins to achieve stable deformation under force action.

During the experiment, one slider is installed on each ring metamaterial of the upper and lower horizontal sides of the periodic homogeneous metamaterials (Supplementary Figs. 39 and 40). The sliders are fixed on the slide rail of the tester to apply quasi-static strain on periodic homogeneous metamaterial. Each ring metamaterials on both vertical sides is equipped with linear bearings, and the linear bearings on one side are connected in series with an optical axis to enhance coordinated deformation of periodic homogeneous metamaterial. The strain rate applied by the testing machine is 20 mm/min. The longitudinal deformation data of periodic homogeneous metamaterial is collected by the control system of the experimental testing machine, and the horizontal deformation data is measured by a ruler. The horizontal strain is the average of the strains at the upper, middle, and lower parts.

## Data availability
All the data supporting the findings of this study are available within the main text and its Supplementary Material. Source data are provided with this paper.

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

## Acknowledgements

We sincerely appreciate the National Natural Science Foundation of China (Grant No. 11902193, Z.Y.) and the Oceanic Interdisciplinary Program of Shanghai Jiao Tong University (Project No. SL2022MS001, Z.Y.) for the financial support to this study.

## Author contributions

X.H., T.T., and Z.Y. conceived the ideas and designed the research. X.H., B.W., and Z.Y. established the theoretical model. X.H. performed the experiments. X.H., T.T., and Z.Y. analyzed the data. X.H., T.T., B.W., and Z.Y. interpreted the results and wrote the manuscript with input from all authors. Z.Y. supervised the study.

## Competing interests

The authors declare no competing interests.
