## [Peer Review File · Nature Communications]

A reprogrammable mechanical metamaterial with origami functional-group transformation and ring reconfigurationReviewers' Comments:

Reviewer #1:

Remarks to the Author:

The authors report a reprogrammable ring metamaterial built by the two origami elements composing each functional group. The reversible structural transformation is possible between the two different functional groups, which provides the multiple functions regarding mechanical properties. This study is well designed and the proposed structures have nice mechanical functions. However, the current manuscript has a lot of inadequate expressions. The reviewer lists some remarks and questions below, that should be addressed.

Major concern:

The function of the metamaterial proposed by the authors is summarized in the statement at Lines 177 to 183. The statement itself is acceptable although there are the inadequate terms (see the minor point #1). Nevertheless, it makes no sense that the analogy of the proposed metamaterial to chemical tautomeric compound as explained in the section of the design concept. Figure 1A illustrates proton tautomerism, in which the two tautomers coexist under a chemical equilibrium state. The tautomerization is caused by proton migration and the geometry of the molecular base framework never changes. On the other hand, the tautomerization called by the authors in this study needs hand manipulations with some kind of fixing components (stoppers) to realize stable geometrical configurations between the |C|R and |R|C elements, as shown in the article figures and the supplementary videos. Since both states are stabilized completely by stoppers, the |C|R and |R|C elements never coexist during the mechanical performances. Therefore, this tautomerization would not be achieved by pure mechanical interaction, which hardly allows readers to understand the advantage of the presented tautomeric metamaterial.

In the current manuscript, tautomeric metamaterial is not scientifically sound. Please carefully ensure the meaning of the double-arrow notation as in Fig. 1B.

Minor points:

1. The authors in the manuscript employ the original terms "physical atoms", "origami atom", "complete-elastic atom" and so on without the explicit definitions or citations. Can "atom" deform? Can "atom" split into two during motion? When tautomerism is explained, the authors raise the real molecular structure consisting of atoms. The ambiguous meanings of "atom" will confuse readers.
2. The authors investigate the Poisson effect in the proposed structures. When a periodic microstructure is considered, it is worth calculating the effective Poisson's ratios under its homogenization. However, this study focuses only on the ring configuration, the mechanical properties of which are not material constants. The reviewer thinks that the developed metamaterials can be extended to periodic structures without loss of their functions. The periodic assembly enables to discuss the effective material constants such as Poisson's ratio.
3. It is well known that stretch-dominated microstructures potentially have negative Poisson's ratio (Rothenburg et al., Nature (1991)). The proposed metamaterials belong to the structures having stretching mechanisms and the auxeticity should be discussed to add the relevant literatures.

Reviewer #2:

Remarks to the Author:

Hu et al present a plethora of what they call tautomeric metamaterials. Overall, the paper seems to be correct, but some parts MUST be improved to meet the standards.

1. First, the abstract is so technical that is not accessible to the general audience. I would recommend the rewriting of the 2nd part.
2. From the first figure, the comparison and the inspiration from panels A, B to C is not so trivial. As

you know in this kind of molecule the stability and the overall interaction are not only given by the 1st order interaction and thus a justification in the geometry constructed is required.

3. In panel C of the fig1, the shrink and dilatation are not consistent with the representation. It lacks the right mechanism.

4. fig1. I do not understand what is displacement axis and it should have some absolute or relative units to link it to the geometry.

5. The caption should be rewritten and more deeply explained as well as the corresponding text in the paper.

6. Figure 2 is done with a very poor quality and is almost impossible to read for some panels. As presented now, all images are useless as I cannot read anything from them. You should deeply reconsider this figure and maybe move some panels to the SI.

7. in fig3. Same as fig 2 but in addition the horizontal axis $\Delta R_i/R_{i0}$ is unclear for different configurations.

8. In panel C, do you have a discontinuity around 0?

9. Fig 4 and 5 are also not clear and way to heavy.

10. Generally here the overall paper should be rewritten, and figure done properly. Math symbols should respect the standards (not bold for scalar, not italic for the names...)

11. Recent literature on triangular and rotating squares should be cited.

Reviewer #1 (Remarks to the Author):

The authors report a reprogrammable ring metamaterial built by the two origami elements composing each functional group. The reversible structural transformation is possible between the two different functional groups, which provides the multiple functions regarding mechanical properties. This study is well designed and the proposed structures have nice mechanical functions. However, the current manuscript has a lot of inadequate expressions. The reviewer lists some remarks and questions below, that should be addressed.

Major concern:

The function of the metamaterial proposed by the authors is summarized in the statement at Lines 177 to 183. The statement itself is acceptable although there are the inadequate terms (see the minor point #1). Nevertheless, it makes no sense that the analogy of the proposed metamaterial to chemical tautomeric compound as explained in the section of the design concept. Figure 1A illustrates proton tautomerism, in which the two tautomers coexist under a chemical equilibrium state. The tautomerization is caused by proton migration and the geometry of the molecular base framework never changes. On the other hand, the tautomerization called by the authors in this study needs hand manipulations with some kind of fixing components (stoppers) to realize stable geometrical configurations between the |C|R and |R|C elements, as shown in the article figures and the supplementary videos. Since both states are stabilized completely by stoppers, the |C|R and |R|C elements never coexist during the mechanical performances. Therefore, this tautomerization would not be achieved by pure mechanical interaction, which hardly allows readers to understand the advantage of the presented tautomeric metamaterial.

In the current manuscript, tautomeric metamaterial is not scientifically sound. Please carefully ensure the meaning of the double-arrow notation as in Fig. 1B.

Minor points:

1. The authors in the manuscript employ the original terms "physical atoms", "origami atom", "complete-elastic atom" and so on without the explicit definitions or citations. Can "atom" deform? Can "atom" split into two during motion? When tautomerism is explained, the authors raise the real molecular structure consisting of atoms. The ambiguous meanings of "atom" will confuse readers.

2. The authors investigate the Poisson effect in the proposed structures. When a periodic microstructure is considered, it is worth calculating the effective Poisson's ratios under its homogenization. However, this study focuses only on the ring configuration, the mechanical properties of which are not material constants. The reviewer thinks that the developed metamaterials can be extended to periodic structures without loss of their functions. The periodic assembly enables to discuss the effective material constants such as Poisson's ratio.

3. It is well known that stretch-dominated microstructures potentially have negative Poisson's ratio (Rothenburg et al., Nature (1991)). The proposed metamaterials belong to the structures having stretching mechanisms and the auxeticity should be discussed to add the relevant literatures.

Reviewer #2 (Remarks to the Author):

Hu et al present a plethora of what they call tautomeric metamaterials. Overall, the paper is seems to be correct, but some parts MUST be improved to meet the standards.

1. First, the abstract is so technical that is not accessible to the general audience. I would recommend the rewriting of the 2nd part.

2. From the first figure, the comparison and the inspiration from panels A, B to C is not so trivial. As you know in this kind of molecule the stability and the overall interaction are not only given by the 1st order interaction and thus a justification in the geometry constructed is required.

3. In panel C of the fig1, the shrink and dilatation are not consistent with the representation. It lacks the right mechanism.

4. fig1. I do not understand what is displacement axis and it should have some absolute or relative units to link it to the geometry.

5. The caption should be rewritten and more deeply explained as well as the corresponding text in the paper.

6. Figure 2 is done with a very poor quality and is almost impossible to read for some panels. As presented now, all images are useless as I cannot read anything from them. You should deeply reconsider this figure and maybe move some panels to the SI.

7. in fig3. Same as fig 2 but in addition the horizontal axis $\Delta R_i/R_i$ is unclear for different configurations.

8. In panel C, do you have a discontinuity around 0?

9. Fig 4 and 5 are also not clear and way to heavy.

10. Generally here the overall paper should be rewritten, and figure done properly. Math symbols should respect the standards (not bold for scalar, not italic for the names...)

11. Recent literature on triangular and rotating squares should be cited.

RESPONSE TO REVIEWER COMMENTS

Reviewer #1

The authors report a reprogrammable ring metamaterial built by the two origami elements composing each functional group. The reversible structural transformation is possible between the two different functional groups, which provides the multiple functions regarding mechanical properties. This study is well designed and the proposed structures have nice mechanical functions. However, the current manuscript has a lot of inadequate expressions. The reviewer lists some remarks and questions below, that should be addressed.

Response: We sincerely appreciate the reviewer for your positive feedback and valuable suggestions regarding our research work. Additionally, we would like to extend our heartfelt gratitude to the reviewer for pointing out the issues pertaining to the manuscript's inadequate expressions. We have provided detailed responses to each suggestion and comment below.

Major concern:

The function of the metamaterial proposed by the authors is summarized in the statement at Lines 177 to 183. The statement itself is acceptable although there are the inadequate terms (see the minor point #1). Nevertheless, it makes no sense that the analogy of the proposed metamaterial to chemical tautomeric compound as explained in the section of the design concept. Figure 1A illustrates proton tautomerism, in which the two tautomers coexist under a chemical equilibrium state. The tautomerization is caused by proton migration and the geometry of the molecular base framework never changes. On the other hand, the tautomerization called by the authors in this study needs hand manipulations with some kind of fixing components (stoppers) to realize stable geometrical configurations between the |C|R and |R|C elements, as shown in the article figures and the supplementary videos. Since both states are stabilized completely by stoppers, the |C|R and |R|C elements never coexist during the mechanical performances. Therefore, this tautomerization would not be achieved by pure mechanical interaction, which hardly allows readers to understand the advantage of the presented tautomeric metamaterial.

In the current manuscript, tautomeric metamaterial is not scientifically sound. Please carefully ensure the meaning of the double-arrow notation as in Fig. 1B.

Response: Thank you to the reviewer for pointing out the issues with the analogy of "tautomerization". During the revision process, we carefully reviewed the definition of tautomeric compound. As the reviewer pointed out, tautomerism is caused by proton migration, and the geometric shape of the molecular base framework remains unchanged. Our designed mechanical metamaterial possesses ring-reconfiguration characteristics, which tautomeric compounds do not have. Regarding the

coexistence of two tautomers, we have addressed this issue by constructing a periodic homogeneous metamaterial, based on the suggestion in minor point #2 by the reviewer. This allows for the coexistence of C-functional (|C|R) ring and R-functional (|R|C) ring within the periodic metamaterial (see video 12). Nevertheless, we still rely on fasteners to achieve stable equilibrium of the rings and their composed periodic homogeneous metamaterial, rather than solely relying on the mechanical performance of the material itself. In contrast, tautomers coexist in spontaneous equilibrium systems and convert into each other at a relatively high rate, which our metamaterial does not accomplish. To summarize, we agree with the reviewer's view that likening our mechanical metamaterial to tautomeric compound is not rigorous. Moreover, emphasizing the transformation of functional groups does not fully convey the advantages of our designed mechanical metamaterial, such as the continuous tunability of positive stiffness, negative stiffness, and negative Poisson's ratio achieved through ring reconfiguration. With the reviewer's comments, we gained a deeper understanding of the essence of the proposed mechanical metamaterial. Therefore, we have made the following modifications throughout the manuscript.

- (1) The title is modified as “*A reprogrammable mechanical metamaterial with origami functional-group transformation and ring reconfiguration* (Page.1, line 1-2)”. Qualitative-change reprogramming of mechanical properties of mechanical metamaterial, such as the switch between complete-elastic (positive stiffness) and rigid-elastic (negative stiffness) responses, is achieved through functional-group transformation. The continuous quantitative-change reprogramming of the mechanical properties, such as continuous tuning of stiffness modulus and Poisson's ratio, is accomplished by ring reconfiguration. The revised title summarizes the reprogrammable features of the proposed mechanical metamaterial via functional group transformation and ring reconfiguration.
- (2) The analogy of the proposed metamaterial to chemical tautomeric compound in the subsection of Design concept is removed. Instead, the design concept of proposed mechanical metamaterials with heterogeneous elements and the comparison with periodic metamaterials with homogenous elements are added in Design concept and Fig. 1a. We add “*The mechanical metamaterials composed of periodically arranged homogeneous elements, such as complete-elastic (C) elements with the positive stiffness and rigid-elastic (R) elements with the negative stiffness, are not mutually transformable (Fig. 1a). This leads to limited reprogramming functions. To enable rich reprogrammable mechanical performances, heterogeneous C and R elements are ingeniously coupled to constitute a C or R functional group. The same type of functional groups is utilized to create the C or R ring metamaterials. The C and R elements can transfer between adjacent functional groups of a ring, allowing for reversible mutual transformation between the C and R ring metamaterials. The ring metamaterial composed of heterogeneous elements can serve as the cell unit to constitute periodic homogeneous metamaterials.*” (Page.2-3, line 56-64) as the first paragraph of Design concept.

Fig. 1 Design concept of reprogrammable mechanical metamaterials with functional-group transformation and ring reconfiguration. **a** Comparison between mechanical metamaterials composed of homogeneous elements and heterogeneous elements: mutually non-transformable behavior is observed within single-element (either red-colored complete-elastic C or blue-colored rigid-elastic R) periodic homogeneous metamaterials. Conversely, the C or R functional group composed of one C and one R elements can mutually transform. Deformation occurs exclusively along the normal direction of the colored solid element sidelines. Design and construction of triangular and quadrilateral ring metamaterial employ alternating heterogeneous C and R elements. Symbols Ω and Γ respectively represent the angular relationship between adjacent elements within ring metamaterials. Ring metamaterials (C ring and R ring) composed of corresponding functional groups can interconvert. Periodic homogeneous metamaterials composed of C or R ring metamaterials can also interconvert, thereby establishing an interchangeability of stress-strain relationships between C and R metamaterials.

- (3) Following Design concept, four subheadings (*Origami heterogeneous elements, Functional member transformation, Ring metamaterial reconfiguration, Periodic homogeneous metamaterial*) were established covering element, functional group, ring metamaterial, and periodic metamaterial according to the architecture level of the proposed metamaterials.
- (4) We have removed the pictorial representation of the analogic molecular formula of the proposed metamaterials in Fig.2-5, and revised all statements about tautomerism in the text, figures, and videos of the main text and supplementary materials.
- (5) All double-arrow notations in figures have been replaced by the customized arrow symbols (arrowhead at either end), shown as the green arrows for functional group transformation and red arrows for ring reconfiguration.

Minor points:

1. The authors in the manuscript employ the original terms "physical atoms", "origami atom", "complete-elastic atom" and so on without the explicit definitions or citations. Can "atom" deform? Can "atom" split into two during motion? When tautomerism is explained, the authors raise the real molecular structure consisting of atoms. The ambiguous meanings of "atom" will confuse readers.

Response: We sincerely thank the reviewer for this rigorous comment. The concept of metamaterials refers to a design methodology for material properties, which involves constructing periodic artificial lattices using "artificial atoms" as the fundamental building elements. The term "artificial atoms" is a metaphorical expression. In the initial manuscript, we used the concept of atoms to represent origami elements as it was meant to draw an analogy with tautomeric compound, but such an expression is inadequate. In the revised version, we have removed the analogy with the tautomerization and rephrased "atom" as the commonly used "element" in material research articles [2, 3] to ensure clarity for the readers. We accordingly modify all the relevant contents in the revised manuscript.

2. Matlack, K.H., Serra-Garcia, M., Palermo, A. *et al.* Designing perturbative metamaterials from discrete models. *Nature Materials* **17**, 323–328 (2018). DOI: 10.1038/s41563-017-0003-3

3. Zhiqiang Meng et al., Deployable mechanical metamaterials with multistep programmable transformation. *Science Advance* **8**, eabn5460(2022). DOI:10.1126/sciadv.abn5460

2. The authors investigate the Poisson effect in the proposed structures. When a periodic microstructure is considered, it is worth calculating the effective Poisson's ratios under its homogenization. However, this study focuses only on the ring configuration, the mechanical properties of which are not material constants. The reviewer thinks that the developed metamaterials can be extended to periodic structures without loss of their functions. The periodic assembly enables to discuss the effective material constants such as Poisson's ratio.

Response: We greatly appreciate the reviewer's constructive suggestions. In the revised manuscript, we have added a subsection titled "Periodic homogeneous metamaterials" to Results. This subsection includes theoretical analyses of the Poisson's ratio for periodic homogeneous metamaterials with cell units of triangular and quadrilateral rings, as well as experimental analyses of the Poisson's ratio for periodic homogeneous metamaterials with quadrilateral rings. Due to shared planar network structure and theoretical foundation, we referred to the research by Rothenburg et al. in *Nature* (1991), and calculated the Poisson's ratio of periodic homogeneous metamaterials with triangular rings to be constant at -1. The inherent properties of periodic homogeneous metamaterials with quadrilateral rings were expressed using the constitutive relationship. Representative volume element was selected as the functional group [R1, R2], and the Poisson's ratio was calculated using the compliance tensor. We theoretically analysed the influence of relevant parameters, such as adjustable inter-group angle and origami stable height, on the Poisson's ratio of periodic homogeneous metamaterials with quadrilateral rings. We fabricated such metamaterials with a 3-row \times 2-column arrays of C rings. For inter-group angles of $\Gamma = 22.5^\circ, 45^\circ, 67.5^\circ, 90^\circ, 112.5^\circ, 145^\circ, \text{ and } 167.5^\circ$, uniaxial compression tests were conducted on C_{III} metamaterials, and uniaxial tension tests were performed on C_I metamaterials. The impact of Γ on the Poisson's ratio and the variation of the Poisson's ratio with strain were analysed.

The added content in **Results** (pages 13-17, lines 315-388) is as follow.

Fig. 6 The auxeticity of periodic homogeneous metamaterials. **a** The paper prototype of periodic homogeneous metamaterials with quadrilateral ring under different configurations (adjustable angles of 22.5°, 45°, 67.5°, 90°, 112.5°, 135° and 157.5°). The geometry of quadrilateral ring can be continuous adjusted and fixed by 3D printed plastic connecting components with detachable pins, see detail in method of Fabrication and experiment testing of periodic homogeneous quadrilateral ring metamaterials. **b** The paper prototype snapshots of an example of periodic homogeneous metamaterial at the strain of 0, 1/3 ϵ , 2/3 ϵ and 3/3 ϵ (from left to right) under compression deformation (example: C_{III} rings with the adjustable angle of 67.5°). The side length l of the physical prototype of paper element is 30 mm. See Methods for the details about periodic homogeneous metamaterial fabrication. **c** The Poisson's ratio of periodic homogeneous metamaterial varied as a function of adjustable angle for compression deformation. The solid line represents theoretical results, and the discrete hollow points denote experimental results. **d** The paper prototype snapshots of an example of periodic homogeneous metamaterial at the strain of 0,

1/3 ϵ , 2/3 ϵ and 3/3 ϵ (from left to right) under tension deformation (example: C_I rings with the adjustable angle of 67.5°). e The Poisson's ratio of periodic homogeneous metamaterial varied as a function of adjustable angle for tension deformation.

Periodic homogeneous metamaterial

The negative Poisson's ratio (auxetic) effect primarily occurs in stretch-dominated structures where the shear stiffness is greater than the axial stiffness²²⁻²⁵, rigid units with relative rotation²⁶⁻²⁸, chiral lattices with node rotation and ligament bending^{24,29}, and sheet-like origami with reoriented structural elements^{28,30}. The R/C functional group constructed in this study belongs to a stretch-dominated structure and achieves the microstructure concept of a shock absorber element with a negative Poisson's ratio, as described in literature 22. In constructing two-dimensional material networks, this functional group, simplified as a spring, can be more easily arranged in various polygon network structures, such as triangles or quadrilaterals, leading to the fabrication of more diverse material structures and properties, compared to re-entrant structures⁵, rotating cube structures³¹⁻³⁴, and Miura origami⁶.

Although the triangular ring metamaterial is difficult to deform under uniaxial force, the metamaterial periodically arranged with triangular ring with a form of planar network is easy to deform under uniaxial tension and compression. The triangular-ring periodic homogeneous metamaterial is considered as a material consisting of randomly packed smooth spherical particles²². Its Poisson's ratio is a constant of -1, related to the stiffness ratio of non-deformable to deformable directions of functional groups³⁵⁻³⁷.

The Poisson's ratio of periodic homogeneous metamaterial of quadrilateral ring is determined by the constitutive relationship when considering the functional group as the representative volume element³⁸⁻⁴⁰. See detail for constitutive relationship in Supplementary Discussions S8c. Since the component of compliance tensor S_{ij} is greater than zero for adjustable angle within (0°, 90°), the periodic homogeneous metamaterials of quadrilateral ring always exhibit auxeticity for this range of adjustable angle³⁸. The Poisson's ratio can be reprogrammed with the element's stable state height H_{stable} and inter-group angle Γ or Ω of the ring configuration. Analytical contour plots show that the Poisson's ratio decreases from 0 to $-\infty$ as the inter-group angle increases and the origami stable height decreases (Fig. S37).

Two prototypes of periodic homogeneous metamaterials composed of quadrilateral C rings were fabricated, see Methods for detail. Instead of replacing fixed supports for the ring metamaterial, 3D printed holes were added at the angle adjustable hinges to secure the ring configuration with pins during the reconfiguring process (Fig. S38, Fig. 6a). The contraction of the metamaterial with quadrilateral C_{III} ring of $\Gamma=67.5^\circ$ under uniaxial compression is observed, as shown in the snapshots in Fig. 6b. The C_{III} elements collapse gradually, while the R elements remain unchanged as the strain increases. Inspection of the internal cavity of the ring metamaterial reveals a gradual decrease in

transverse size, demonstrating its auxetic characteristics. Similar behaviours are observed for periodic homogeneous metamaterials with larger Γ (Supplementary Video 10). The experimental and theoretical results show a consistent trend of reprogrammable changes (Fig. 6c). The Poisson's ratio of the periodic homogeneous metamaterial with quadrilateral C_{III} ring decreases exponentially from -0.336 to -1.717 with the increase of Γ from 22.5° to 157.5°, indicating more pronounced auxetic characteristics for larger Γ . The experimental results also show slight fluctuations in the Poisson's ratio during compression.

Fig. 6d and Supplementary Video 11 display the expansion of periodic homogeneous metamaterial consisting of quadrilateral C_I rings under axial tension. The synchronous increase in longitudinal and transverse dimensions indicates its auxeticity characteristics. Similar to the contraction of auxetic behaviour, the Poisson's ratio exponentially decreases as Γ increases (Fig. 6e). The theoretical and experimental results for the expansion of auxetic behaviour align more closely compared to the results for the contraction of auxetic behaviour. This is attributed to the more uniform unfolding process of C_I elements in the metamaterial, compared to the folding process of C_{III} . By varying the inter-group angle from 22.5° to 157.5°, the Poisson's ratio can be reprogrammed from -0.219 to -1.853.

The published tuning of negative Poisson's ratio typically occurs during structural deformation, often accompanied by structural reconfiguration^{5,6}, making it difficult to maintain a stable configuration under large deformations. In contrast, our experimental validation demonstrates that the negative Poisson's ratio effect persists in stable configurations and can be controlled by adjusting the angles between functional groups, allowing for modifications in the stable configuration and its auxetic behaviour. Additionally, the rings in the periodic homogeneous metamaterials can independently switch between C and R functional modes without mutual interference. The coexistence of C rings and R rings (see Supplementary Video 12) can also impact the mechanical performance of the metamaterial. These advantages render this metamaterial suitable for various applications, including load-bearing, soft actuation, and mechanical computing.

*The added content in **Methods** (Pages 23-24, lines 559-594) is as follow.*

Fabrication and experiment testing of periodic homogeneous quadrilateral ring metamaterials. The reprogrammable auxeticity of quadrilateral ring metamaterials are further investigated with a periodic homogeneous form and paper prototype elements. To ensure the effective use of the experimental space and the better folding of the elements, we set the size of the paper prototype element to 30mm × 30mm × 30mm (length × width × height). And the periodic homogeneous metamaterial is assembled by six ring metamaterials with an array of three rows and two columns (Fig. S38). The paper prototype of element is made by folding red and blue cardboard with pre made creases. The pre made creases on cardboard are cut by cutting plotter. The red paper prototypes represent C elements, and the blue paper prototypes represent R elements. The standard of the cardboard is 180 g/m².

The paper prototype is equipped with two 3D printed square frames at both ends to fix the shape.

There are circular holes with a diameter of 0.8 mm on each side of square frame (Fig. S38a). Among them, one set of holes on the opposite sides is used for fixing the square frame and the paper prototype with screws of M1×3 mm, and one set of holes on the other opposite sides is used for installing connecting components between elements with screws of M1×4 mm. There are two types of connecting components, the first of which is installed on the square frames of C elements. The second ones are installed on that of R elements. The circular plates on one corner of first type of connecting components and the circular plate on one corner of the second type of connecting components overlap and are connected by screws to form a rotating hinge (Fig. S38b). Based on this method, the functional group is constructed by one C paper prototype and one R paper prototype. According to the aforementioned functional group connection, the ring metamaterial with functional group transformation is completed (Fig. S38c). Circular plates are left on each corner of ring metamaterial to facilitate the connection for constructing periodic homogeneous metamaterial. Sixteen circle holes with a diameter of 0.7 mm and the interval of 22.5° is reserved on lower circular plate of each hinge. Two circle holes with a diameter of 0.7 mm and the interval of 180° is reserved on upper circular plate of each hinge. The circle hole of the upper coincide different circle hole on lower circular plates to adjust the internal angle of the ring metamaterial. These coinciding holes can be fixed by pins to achieve stable deformation under force action.

During experiment, one slider is installed on each ring metamaterial of the upper and lower horizontal sides of the periodic homogeneous metamaterials (Fig. S39 and S40). The sliders are fixed on the slide rail of the tester to apply quasi-static strain on periodic homogeneous metamaterial. Each ring metamaterials on both vertical sides is equipped with linear bearings, and the linear bearings on one side are connected in series with an optical axis to enhance coordinated deformation of periodic homogeneous metamaterial. The strain rate applied by the testing machine is 20 mm/min. The longitudinal deformation data of periodic homogeneous metamaterial is collected by the control system of the experimental testing machine, and the horizontal deformation data is measured by a ruler. The horizontal strain is the average of the strains at the upper, middle, and lower parts.

*The revised and added content in **Supplementary material** is as follows.*

Supplementary Fig. 24: representative volume element of periodic homogeneous quadrilateral ring metamaterials.

For auxetic deformation for reprogrammable quadrilateral ring metamaterials as shown in Fig.4 and Fig.5, origami elements with the same mechanical response deform cooperatively. Although this Metamaterial has strong nonlinear mechanical properties, its mechanical response under small deformation can be considered as linear. In addition, judging from the structural deformation, the Poisson's ratio of the Metamaterial is constant at a fixed adjustment angle, so we can calculate the material properties, such as Poisson's ratio, of the Metamaterial under small structural deformation. Therefore, we can use the constitutive relationship to reflect the inherent properties of metamaterials. For the metamaterial in this article, the constitutive relationship can be expressed as

$$\varepsilon_i = S_{ii}\sigma_i \Rightarrow \begin{Bmatrix} \varepsilon_1 \\ \varepsilon_2 \end{Bmatrix} = \begin{bmatrix} S_{11} & S_{12} \\ sym & S_{22} \end{bmatrix} \begin{Bmatrix} \sigma_1 \\ \sigma_2 \end{Bmatrix} \quad (S48)$$

ε and σ are strain and stress of metamaterials, respectively. $i=1,2$, which represents the a and t directions, respectively, as shown in Fig. S24. S is the compliance tensor. The functional-group is considered as representative volume element. The length of representative volume element at a direction is calculated by

$$L_a = (H_{\max} + H) \cos\left(\frac{A}{2}\right) + H_{\max} \sin \frac{A}{2} \quad (S49)$$

where A is the internal angle of reprogrammable mechanical metamaterials with quadrilateral ring. $A = \Gamma$ when reprogrammable mechanical metamaterial is transformed to C ring metamaterials, and $A = \Omega$ when reprogrammable mechanical metamaterial is transformed to R ring metamaterials. The length of representative volume element at t direction is calculated by

$$L_t = (H_{\max} + H) \sin\left(\frac{A}{2}\right) + H_{\max} \cos \frac{A}{2} \quad (S50)$$

When the metamaterial undergoes deformation, the boundary conditions of the representative volume element are

$$\begin{cases} \sigma_1 = F \cos\left(\frac{A}{2}\right) / (H_{\max} L_t) \\ \sigma_2 = F \sin\left(\frac{A}{2}\right) / (H_{\max} L_a) \end{cases} \quad (S51)$$

The concentrated force of metamaterial is

$$\begin{cases} F_a = mF \cos\left(\frac{A}{2}\right) \\ F_t = nF \sin\left(\frac{A}{2}\right) \end{cases} \quad (S52)$$

where m and n are the number of representative volume element at t and a direction, respectively. Similarly, the engineering stress is calculated as

$$\begin{cases} \sigma_a = mF \cos\left(\frac{A}{2}\right) / (H_{\max} L_t) \\ \sigma_t = nF \sin\left(\frac{A}{2}\right) / (H_{\max} L_a) \end{cases} \quad (S53)$$

Assuming that the deformation of the representative volume element is δl , the Strain energy density of representative volume element is determined as

$$U = k\delta l^2 / 2V \quad (S54)$$

where k is the elements' stiffness under small deformation. V is the volume in which the potential energy U is considered. The components of the compliance tensor can be calculated as

$$S_{11} = \frac{2U}{\sigma_1 \sigma_1}; \quad S_{12} = S_{21} = \frac{2U}{\sigma_1 \sigma_2}; \quad S_{22} = \frac{2U}{\sigma_2 \sigma_2} \quad (S55)$$

In addition, the components of compliance tensor can also be represented as

$$S_{11} = \frac{1}{E_1}; \quad S_{12} = S_{21} = -\frac{\nu_{12}}{E_1} = -\frac{\nu_{21}}{E_2}; \quad S_{22} = \frac{1}{E_2} \quad (S56)$$

Thus, the Poisson's ratios are defined as

$$\nu_{ta} = -\frac{\sin(A/2) \left[H_{\max} (\cos(A/2) + \sin(A/2)) + H_{stable} \cos(A/2) \right]}{\cos(A/2) \left[H_{\max} (\cos(A/2) + \sin(A/2)) + H_{stable} \sin(A/2) \right]} \quad (S57)$$

Supplementary Fig. 38: The fabrication of periodic homogeneous metamaterials with quadrilateral

rings. (a) The fabricated paper prototype of origami elements equipped with square frames, connecting components and their presentations from different perspectives. (b) The fabricated prototype of complete elastic functional-group. (c) The fabricated prototype of complete elastic ring metamaterial, and the implementation method of angle adjustment and fixation under deformation. (d) The fabricated of periodic homogeneous metamaterials with quadrilateral rings.

Supplementary Fig. 39: Uniaxial mechanical tests of auxeticity under compression for periodic homogeneous metamaterials with C_{III} rings. The adjusting angle is fixed with 22.5° , 45° , 67.5° , 90° , 112.5° , 135° and 157.5° .

Supplementary Fig. 40: Uniaxial mechanical tests of auxeticity under tension for periodic homogeneous metamaterials with C_{III} rings. The adjusting angle is fixed with 22.5°, 45°, 67.5°, 90°, 112.5°, 135° and 157.5°.

Supplementary Fig. 41: The periodic homogeneous metamaterials with C_{III} cell units and transformed one with R cell units. (a) The periodic homogeneous metamaterials with C_{III} and R cell units. (b) The connecting components of periodic homogeneous metamaterials between origami elements. In order to improve the transformation efficiency of Metamaterial, the connecting components are provided with sunken holes and protruding blocks that can be fastened with each other to replace the fixation of screws.

3. It is well known that stretch-dominated microstructures potentially have negative Poisson's ratio (Rothenburg et al., Nature (1991)). The proposed metamaterials belong to the structures having stretching mechanisms and the auxeticity should be discussed to add the relevant literatures.

Response: We would like to express our sincere gratitude to the reviewer for providing us with the suggestion of this reference examining the mechanisms behind the generation of negative Poisson's

ratio. We have thoroughly studied this reference and have also conducted an extensive search for review and research papers on auxetic behaviour. The mechanisms that have been mainly proposed to achieve a negative Poisson's ratio (auxetic behaviour) in current research are as follows:

1. The first mechanism is the stretching mechanism mentioned by the reviewer. In the study by Rothenburg et al., *Nature* (1991) (22), it is mentioned that the stiffness in axial compression or tension is much smaller than the shear stiffness of a shock absorber element. The triangular structures composed of shock absorber elements form an isotropic structure (with shock absorber elements present in all directions with equal probability), showing a negative Poisson's ratio of -1. The same mechanism applies to the Re-entrant structure mentioned in Rothenburg et al., *Nature* (1991), where the axial stiffness is low, forming a two-dimensional honeycomb (23, 24) or three-dimensional foam-like structure (25) that permanently collapses inward during compression, exhibiting a negative Poisson's ratio. Both structures share the same mechanism for achieving a negative Poisson's ratio: the stiffness of microstructural elements during shear is larger than that during compression, as referred to by the reviewer as "stretch-dominated." Our metamaterials also belong to this mechanism.

2. The second mechanism involves structures composed of rotating rigid units (26, 27, 28), which generate the auxetic effect through the relative rotation of these rigid units (rotating hinged elements) during deformation.

3. The third mechanism involves chiral lattices (24, 29), where the auxetic behaviour is caused by the rotation of lattice nodes and the bending of ligaments.

4. The fourth mechanism is seen in sheet-like or layered structures, such as Miura origami, which exhibit a negative Poisson's ratio through the reorientation of structural elements during the folding/unfolding process (28, 30).

Currently, the adjustment of negative Poisson's ratio typically occurs during the deformation process of the structure (5, 6), where the structure undergoes reconstruction. In contrast, the negative Poisson's ratio of our metamaterials is experimentally obtained in a stable configuration, and can be adjusted by changing the angles between the functional groups to alter the stable configuration and modulate its auxetic behaviour.

5. Farzaneh, A., Pawar, N., Portela, C. M., & Hopkins, J. B. Sequential metamaterials with alternating Poisson's ratios. *Nature Communication* **13**, 1041 (2022). DOI: 10.1038/s41467-022-28696-9

6. Pratapa, P. P., Liu, K., & Paulino, G. H. Geometric mechanics of origami patterns exhibiting Poisson's ratio switch by breaking mountain and valley assignment. *Physical review letters* **122**, 15: 155501 (2019). DOI: 10.1103/PhysRevLett.122.155501

22. Rothenburg L, Ai. Berlin A, & Bathurst R J. Microstructure of isotropic materials with negative Poisson's ratio. *Nature* **354**(6353):470-472 (1991). DOI:10.1038/354470a0.

23. Lakes R . Advances in negative Poisson's ratio materials. *Advanced Materials* **5**(4):293-296 (2010). DOI:10.1002/adma.19930050416.

24. Lakes R. Deformation mechanisms in negative Poisson's ratio materials: structural aspects. *Journal of materials science* **26**: 2287-2292 (1991). DOI:10.1007/bf01130170
25. Lakes R. Foam structures with a negative Poisson's ratio. *Science* **235**(4792): 1038-1040 (1987). DOI:10.1126/science.235.4792.1038
26. Grima J N, & Evans K E. Auxetic behavior from rotating squares. *Journal of materials science letters* **19**: 1563-1565 (2000). DOI:10.1023/A:1006781224002
27. Grima J N, Alderson A, & Evans K E. Auxetic behaviour from rotating rigid units. *PHYSICA STATUS SOLIDI B-BASIC SOLID STATE PHYSICS* **242**(3): 561-575 (2005). DOI:10.1002/pssb.200460376
28. Lakes R S. Negative-Poisson's-ratio materials: auxetic solids. *Annual review of materials research* **47**: 63-81 (2017). DOI:10.1146/annurev-matsci-070616-124118
29. Prall D, & Lakes R S. Properties of a chiral honeycomb with a Poisson's ratio of -1. *International Journal of Mechanical Sciences* **39**(3): 305-314 (1997). DOI:10.1016/S0020-7403(96)00025-2
30. Wei Z Y., Guo, ZV., Dudte, L., Liang, HY., & Mahadevan, L. Geometric mechanics of periodic pleated origami. *Physical review letters* **110**(21): 215501 (2013). DOI:10.1103/PhysRevLett.110.215501

We add the discussion of auxeticity with the relevant literatures in Results, as follows.

(Page.13-14, line 316-325) The negative Poisson's ratio (auxetic) effect primarily occurs in stretch-dominated structures where the shear stiffness is greater than the axial stiffness²²⁻²⁵, rigid units with relative rotation²⁶⁻²⁸, chiral lattices with node rotation and ligament bending^{24,29}, and sheet-like origami with reoriented structural elements^{28,30}. The R/C functional group constructed in this study belongs to a stretch-dominated structure and achieves the microstructure concept of a shock absorber element with a negative Poisson's ratio, as described in literature 22. In constructing two-dimensional material networks, this functional group, simplified as a spring, can be more easily arranged in various polygon network structures, such as triangles or quadrilaterals, leading to the fabrication of more diverse material structures and properties, compared to re-entrant structures⁵, rotating cube structures³¹⁻³⁴, and Miura origami⁶.

(Page.16-17, line 379-388) The published tuning of negative Poisson's ratio typically occurs during structural deformation, often accompanied by structural reconfiguration^{5,6}, making it difficult to maintain a stable configuration under large deformations. In contrast, our experimental validation demonstrates that the negative Poisson's ratio effect persists in stable configurations and can be controlled by adjusting the angles between functional groups, allowing for modifications in the stable configuration and its auxetic behaviour. Additionally, the rings in the periodic homogeneous metamaterials can independently switch between C and R functional modes without mutual interference. The coexistence of C rings and R rings (see Supplementary Video 12) can also impact the mechanical performance of the metamaterial. These advantages render this metamaterial suitable for various applications, including load-bearing, soft actuation, and mechanical computing.

Reviewer #2:

Hu et al present a plethora of what they call tautomeric metamaterials. Overall, the paper is seems to be correct, but some parts MUST be improved to meet the standards.

Response: Thank you sincerely for the recognition of our research work, and we would also like to express our deep appreciation for the reviewer's comprehensive review of our paper, including the standards for writing and graphic expression. We have taken each comment into careful consideration and have responded to them individually. Please find our responses below.

1. First, the abstract is so technical that is not accessible to the general audience. I would recommend the rewriting of the 2nd part.

Response: We appreciate the valuable comment from the reviewer. In the original version, the second part of the abstract presented a comprehensive summary of the technical aspects, which might have been vague and difficult to understand. In the revised version, we have provided a detailed description of the significant outcomes of our study on metamaterials at four hierarchical levels: origami elements, functional groups, ring metamaterials, and periodic metamaterials. This modification aims to enhance the clarity of the research work, making it more accessible for a general audience to comprehend.

The revised abstract (Page.1, line 16-26):

Here, we introduce a reprogrammable mechanical metamaterial composed of origami elements with heterogeneous mechanical properties, which achieves various mechanical behavior patterns by functional group transformations and ring reconfigurations. Through the anisotropic assembly of two heterogeneous elements into a functional group, we enable mechanical behavior switching between positive and negative stiffness. The resulting polygonal ring exhibits rotational deformation, zero Poisson's ratio stretching/compression deformation, and negative Poisson's ratio auxetic deformation. Arranging these rings periodically yields homogeneous metamaterials. The reconfiguration of quadrilateral rings allows for continuous fine-tunability of the mechanical response and negative Poisson's ratio. This mechanical metamaterial provides a versatile material platform for reprogrammable mechanical computing, multi-purpose robots, transformable vehicles and architectures at different scales.

2. From the first figure, the comparison and the inspiration from panels A, B to C is not so trivial. As you know in this kind of molecule the stability and the overall interaction are not only given by the 1st order interaction and thus a justification in the geometry constructed is required.

Response: We sincerely appreciate the reviewer's professional comment on the analogy to tautomeric compounds. We acknowledge the concerns raised in this regard. Firstly, compounds achieve spontaneous stable equilibrium through covalent and secondary bonding, whereas our ring metamaterials require external forces to tighten fasteners to ensure the stability of the ring and periodic metamaterial structure. Relying solely on the fitting between the elements and hinge connection between functional groups is not enough to ensure stable transmission of stress between elements at the axial direction of the functional group. This may cause geometric changes of the ring configuration. Thus, the structural supports, as shown in the blue boxes of the Fig. 1, are installed between functional groups and elements to maintain geometry in the mechanical performance testing. We notice that the replacement of structural support increases the reconfiguration difficulty of metamaterial constructed by more functional groups. Thus, we improved the angle fixing method. Instead of replacing the fixed supports, the prototype of periodic homogeneous metamaterial is equipped with holes at the angle adjustment hinges to fix ring configuration with pins after the reconfiguring process, as shown in the green boxes of fig. 2 and added description in Methods. This method greatly optimizes the reprogrammable process of reconfigurable metamaterial. From the demonstration videos on deformation, metamaterial deforms while the angles are basically unchanged after fixation.

Fig. 1. The prototype snapshots of quadrilateral ring metamaterial with the structural supports (in blue boxes).

Fig. 2. The fabricated prototype of complete elastic ring metamaterial, the implementation method of continuous angle adjustment and fixation under deformation, periodic homogeneous metamaterials with quadrilateral rings.

We add the illustrations in the geometry in main text as follow

(Page.8, line 198-199) The geometry of triangular ring metamaterials is fixed by 3D printed plastic structural supports, see detail in Fig. S27.

(Page.11, line 265-267) The geometry of quadrilateral ring metamaterials is fixed by 3D printed plastic structural supports, see detail in Fig. S27.

(Page.14-15, line 329-331) The geometry of quadrilateral ring can be continuous adjusted and fixed by 3D printed plastic connecting components with detachable pins, see detail in method of Fabrication and experiment testing of periodic homogeneous quadrilateral ring metamaterials.

(Page.22, line 529-530) The remaining narrow sides are arranged with outwardly convex cylinders to install structural supports and fix the angle of the reprogrammable mechanical metamaterials.

(Page.24, line 581-585) Sixteen circle holes with a diameter of 0.7 mm and the interval of 22.5° is reserved on lower circular plate of each hinge. Two circle holes with a diameter of 0.7 mm and the interval of 180° is reserved on upper circular plate of each hinge. The circle hole of the upper coincide different circle hole on lower circular plates to adjust the internal angle of the ring metamaterial. These coinciding holes can be fixed by pins to achieve stable deformation under force action.

Although the connection shown as the pins within and between the ring unit structure of the periodic metamaterial can be regarded as first-order and second-order interactions, respectively, our quadrilateral ring currently relies on external force tightening to achieve stable configurations and does not exhibit inherent mechanical stability. The quadrilateral ring structure can reconfigure, while the molecular geometry of the compounds remains unchanged. Therefore, taking into consideration the characteristics of our metamaterials and adopting a rigorous attitude towards the analogy of tautomeric compounds, we have decided to remove the analogy to the compounds. The title is modified as “A reprogrammable mechanical metamaterial with origami functional-group transformation and ring reconfiguration (Page.1, line 1-2)”. We focus on summarizing the design concept of our reprogrammable metamaterial, which involves the successful integration of functional groups through the assembly of heterogeneous origami elements. By transforming these functional groups, we achieve a complete change in mechanical behaviour and enable continuous tuning of mechanical properties through ring reconfigurations. This design concept is distinct from existing reprogrammable metamaterials with homogeneous elements, as it allows for both qualitative changes (complete-elastic response with positive stiffness throughout overall length and rigid-elastic response with an initially rigid section followed by a negative stiffness elastic section) and quantitative variations (continuous adjustments of positive/negative stiffness and Poisson's ratio), please see the revised Figure 1 (Page.3, line 65-76).

Fig. 1 Design concept of reprogrammable mechanical metamaterials with functional-group transformation and ring reconfiguration. a Comparison between mechanical metamaterials composed of homogeneous elements and heterogeneous elements: mutually non-transformable behavior is observed within single-element (either red-colored complete-elastic C or blue-colored rigid-elastic R) periodic homogeneous metamaterials. Conversely, the C or R functional group composed of one C and one R elements can mutually transform. Deformation occurs exclusively

along the normal direction of the colored solid element sidelines. Design and construction of triangular and quadrilateral ring metamaterial employ alternating heterogeneous C and R elements. Symbols Ω and Γ respectively represent the angular relationship between adjacent elements within ring metamaterials. Ring metamaterials (C ring and R ring) composed of corresponding functional groups can interconvert. Periodic homogeneous metamaterials composed of C or R ring metamaterials can also interconvert, thereby establishing an interchangeability of stress-strain relationships between C and R metamaterials.

We rewrote the Design concept regarding Fig. 1a (Pages 2-4, lines 56-90) as follows.

The mechanical metamaterials composed of periodically arranged homogeneous elements, such as complete-elastic (C) elements with the positive stiffness and rigid-elastic (R) elements with the negative stiffness, are not mutually transformable (Fig. 1a). This leads to limited reprogramming functions. To enable rich reprogrammable mechanical performances, heterogeneous C and R elements are ingeniously coupled to constitute a C or R functional group. The same type of functional groups is utilized to create the C or R ring metamaterials. The C and R elements can transfer between adjacent functional groups of a ring, allowing for reversible mutual transformation between the C and R ring metamaterials. The ring metamaterial composed of heterogeneous elements can serve as the cell unit to constitute periodic homogeneous metamaterials.

The size of the periodic unit structure, i.e., the ring metamaterial, can be minimized to three. The geometry of the triangular ring metamaterial exhibits triangular stability parameters of $\Omega = 0^\circ$ and $\Gamma = 60^\circ$ for the C ring metamaterial or $\Omega = 60^\circ$ and $\Gamma = 0^\circ$ for the R ring metamaterial. Meanwhile, for the quadrilateral ring metamaterial, its geometry can be adjusted based on the parameter angle Γ for the C ring metamaterial or Ω for the R ring metamaterial (where $0 \leq \Omega/\Gamma \leq 180^\circ$) due to its parallelogram structural features, offering more reconfiguration and greater reprogrammability than the triangular ring.

3. In panel C of the fig1, the shrink and dilatation are not consistent with the representation. It lacks the right mechanism.

Response: We sincerely thank the reviewer for this insightful comment. In accordance with the recommendations from the reviewer, we have differentiated three distinct mechanisms for contraction and expansion, i.e., those induced by rotation, uniaxial deformation with a Poisson's ratio of zero, and auxetic deformation with a negative Poisson's ratio. These distinctions were made to accurately depict the deformation mechanisms in Fig. 1b (previously labelled as Fig. 1c) with modified figure caption (Page 3, lines 76-80). Additionally, we have utilized consistent colour for both the text and the corresponding deformation image in Figure 1b to avoid any potential confusion in visual representation.

Fig. 1 b The schematic of multimode deformation of designed mechanical metamaterials with an example of quadrilateral ring. Contraction and expansion deformations are shown in dark green and dark yellow blocks, respectively. The rotational, uniaxial (zero Poisson's ratio), auxetic (negative Poisson's ratio) contraction and expansion shown in the same row have the same force action mode but opposite direction (black arrows).

We add the description in Design concept (Page.4, line 91-95):

The quadrilateral ring metamaterial with both rotational and axial symmetries was designed to exhibit torsional deformation, axial deformation with zero Poisson's ratio and auxetic deformation with negative Poisson's ratio (Fig. 1b). The ring's multimode deformation is facilitated by the compressible/tensile C elements and compressible R elements, allowing it to undergo contraction, expansion, or both.

4. fig1. I do not understand what is displacement axis and it should have some absolute or relative units to link it to the geometry.

Response: We greatly appreciate the comment and suggestion provided by the reviewer. The force-displacement curve depicted in Figure 1 is not a quantitative result derived from theoretical calculations or experimental testing, but rather a qualitative illustration intended to compare two distinct mechanical behaviours. To avoid any potential confusion for the readers, we have replaced the force-displacement curve with the stress-strain curve for R and C metamaterials for qualitative illustration. Strain is a dimensionless quantity that represents deformation relative to the original length. Meanwhile, we add the unit of the stress axis on the figure. The modified content is shown in fig. 1.

See fig. 1a under answer to question 2 and fig. 1c as below.

Fig. 1 c The schematic of quadrilateral ring reconfiguration (example: R ring metamaterial with $\Omega = 0^\circ, 60^\circ, 90^\circ, 120^\circ$ and 180°) and the continuously reprogrammable mechanical responses for C and R metamaterials by ring reconfigurations.

We modified the relevant description in Design concept (page 4, lines 95-99).

The geometry of the quadrilateral ring is able to transform topologically from square to rhombus and parallel lines, referred as the ring reconfiguration. Taking the R ring metamaterial for instance, its parameter angle Ω is tuned within the ring reconfiguration range of $[0^\circ, 180^\circ]$ (Fig. 1c). Through the ring reconfigurations, nonlinear stress-strain curves of both the C and R metamaterials, respectively having positive and negative stiffness, can be continuously reprogrammed.

5. The caption should be rewritten and more deeply explained as well as the corresponding text in the paper.

Response: Thank you very much for your professional suggestion. We have rewritten the caption in order to convey the information contained in the figure more clearly. The revised content about caption of Fig. 1 is as follow (Page.3, line 65-82).

Fig. 1 Design concept of reprogrammable mechanical metamaterials with functional-group transformation and ring reconfiguration. **a** Comparison between mechanical metamaterials composed of homogeneous elements and heterogeneous elements: mutually non-transformable behavior is observed within single-element (either red-colored complete-elastic C or blue-colored rigid-elastic R) periodic homogeneous metamaterials. Conversely, the C or R functional group composed of one C and one R elements can mutually transform. Deformation occurs exclusively along the normal direction of the colored solid element sidelines. Design and construction of triangular and quadrilateral ring metamaterial employ alternating heterogeneous C and R elements. Symbols Ω and Γ respectively represent the angular relationship between adjacent elements within ring metamaterials. Ring metamaterials (C ring and R ring) composed of corresponding functional groups can interconvert. Periodic homogeneous metamaterials composed of C or R ring metamaterials can also interconvert, thereby establishing an interchangeability of stress-strain relationships between C and R metamaterials. **b** The schematic of multimode deformation of designed mechanical metamaterials with an example of quadrilateral ring. Contraction and expansion deformations are shown in dark green and dark yellow blocks, respectively. The rotational, uniaxial (zero Poisson's ratio), auxetic (negative Poisson's ratio) contraction and expansion shown in the same row have the same force action mode but opposite direction (black arrows). **c** The schematic of quadrilateral ring reconfiguration (example: R ring metamaterial with $\Omega = 0^\circ, 60^\circ, 90^\circ, 120^\circ$ and 180°) and the continuously reprogrammable mechanical responses for C and R metamaterials by ring reconfigurations.

We have also included more deeply textual description on Fig. 1 in the section of Design concept (pages2-4, lines 55-99).

Design concept

The mechanical metamaterials composed of periodically arranged homogeneous elements, such as complete-elastic (C) elements with the positive stiffness and rigid-elastic (R) elements with the negative stiffness, are not mutually transformable (Fig. 1a). This leads to limited reprogramming functions. To enable rich reprogrammable mechanical performances, heterogeneous C and R elements are ingeniously coupled to constitute a C or R functional group. The same type of functional groups is utilized to create the C or R ring metamaterials. The C and R elements can transfer between adjacent functional groups of a ring, allowing for reversible mutual transformation between the C and R ring metamaterials. The ring metamaterial composed of heterogeneous elements can serve as the cell unit to constitute periodic homogeneous metamaterials.

The size of the periodic unit structure, i.e., the ring metamaterial, can be minimized to three. The geometry of the triangular ring metamaterial exhibits triangular stability parameters of $\Omega = 0^\circ$ and $\Gamma = 60^\circ$ for the C ring metamaterial or $\Omega = 60^\circ$ and $\Gamma = 0^\circ$ for the R ring metamaterial. Meanwhile, for the quadrilateral ring metamaterial, its geometry can be adjusted based on the parameter angle Γ for the C ring metamaterial or Ω for the R ring metamaterial (where $0 \leq \Omega/\Gamma \leq 180^\circ$) due to its parallelogram structural features, offering more reconfiguration and greater reprogrammability than the triangular ring.

The quadrilateral ring metamaterial with both rotational and axial symmetries was designed to exhibit torsional deformation, axial deformation with zero Poisson's ratio and auxetic deformation with negative Poisson's ratio (Fig. 1b). The ring's multimode deformation is facilitated by the compressible/tensile C elements and compressible R elements, allowing it to undergo contraction, expansion, or both. The geometry of the quadrilateral ring is able to transform topologically from square to rhombus and parallel lines, referred as the ring reconfiguration. Taking the R ring metamaterial for instance, its parameter angle Ω is tuned within the ring reconfiguration range of $[0^\circ, 180^\circ]$ (Fig. 1c). Through the ring reconfigurations, nonlinear stress-strain curves of both the C and R metamaterials, respectively having positive and negative stiffness, can be continuously reprogrammed.

6. Figure 2 is done with a very poor quality and is almost impossible to read for some panels. As presented now, all images are useless as I cannot read anything from them. You should deeply reconsider this figure and maybe move some panels to the SI.

***Response:** Thank you very much for the figure suggestions. We have carefully considered the feedback and agree that the content in Figure 2 of the first submission was indeed excessive. After careful deliberation, we have identified the most important and core aspects of the figure, which are the design of origami elements with different mechanical properties and the explanation of functional groups formed by the combination of two different origami elements. Thus, we have retained the crease pattern, examples of the expansion and contraction effects of dihedral angles, physical diagrams of the four origami elements along with their experimental and theoretical results on stress-strain*

relationship, and physical diagrams of the four functional groups. We have moved the physical description of the facet I, II and III in panel a, crease mechanics of panel b, physical realization of crease in panel c and origami element design in panel d to the supplementary material. The transformation representation of functional groups in panel e has also been removed because it is able to show with demonstration video.

Additionally, we have rewritten the caption for Figure 2. The picture brief introduction of origami base, crease pattern, its 3D structure, origami facets, origami creases and 3D dimensional parameters are added for panel a in caption. The picture brief introduction of crease expansion for current dihedral angle large than initial dihedral angle and crease contraction for current dihedral angle less than initial dihedral angle, and facet fabrication are added for panel b. The picture brief introduction of deformation models for C_I, C_{II}, C_{III} and R origami elements, experiment results and origami prototype assembly manufacturing is added for panel c. The picture brief introduction of constructed functional groups of C_I, C_{II}, C_{III} and R is added for panel d. We have also rewritten the corresponding texts, please see the revised manuscript (page 5-6 line 112-129).

Fig. 2 The designed origami element and constructed functional group. **a** The crease pattern and 3D schematic of designed origami element. The base with three classes of facets is displayed in the left red dashed box. Each kind of creases are displayed in the right red dashed box. The black arrow points the axial direction of the element. Symbols φ_i represent the independent angle parameter of the origami element, and i corresponds to the crease 1, 2, 3 and 4. **b** The physical demonstration of expansion and contraction tendency of crease segments under different initial dihedrals (example: crease 4). The facets with high stiffness are manufactured using photosensitive resin by light curing three-dimensional (3D) printing technique for mechanical testing. Creases with representative angles are produced through fused deposition modeling 3D printing technique of thermoplastic polyurethane (TPU) with a hardness of 95A. See Methods for the details about origami element fabrication. **c** The snapshots and mechanical responses of the C_I , C_{II} , C_{III} and R elements. The deformable modes of origami elements from left to right are full-length tension (C_I), partial-length tension and compression (C_{II}), full-length compression (C_{III}) and full-length compression with a static bifurcation behavior (R). The snapshots display the physical prototypes of origami element in stable state and maximum deformation state. The selected initial dihedral angle for creases of the physical prototype is list in Supplementary Fig. S13(b) and the side length l of the physical prototype is 110 mm. The error bars present the error burst for multiple experimental subjects testing. **d** The prototype demonstration and transformation schematic (green arrows) of C_I , C_{II} , C_{III} and R functional group constructed by one C (C_I , C_{II} or C_{III}) element and one R element shown in panel c. The white arrow indicates the deformable direction of functional group.

7. in fig3. Same as fig 2 but in addition the horizontal axis $\Delta R_i/R_i$ is unclear for different configurations.

Response: Thank you for your comment, and we fully agree that a better panel arrangement should be given for Fig. 3. We use Fig. 3 to mainly express the torsional deformation of the designed metamaterial with rotational symmetry and the mechanical properties under this deformation. The ring metamaterials with C_I functional groups produces torsional expansion deformation when torque is applied. The ring metamaterials with C_{III} and R functional groups produces torsional contraction deformation. And the ring metamaterials with C_{II} functional groups produces both torsional expansion and contraction deformation. Because the torsional deformations of triangular and quadrilateral rings are similar, we finally demonstrate the torsional deformation of the triangular ring metamaterial in panel A. The physical demonstration images about torsional deformation of the quadrilateral ring are moved to supplementary materials. The original data graphs are preserved in panels b-d, as follows. We have also rewritten the corresponding texts, please see the revised manuscript (page 8-9, lines 194-208).

Fig. 3 The torsional deformation and mechanical characterization of triangular ring metamaterials. **a** The prototype snapshots of initial stable state (leftmost column) and the final deformed state (rightmost column) of designed triangular ring metamaterials. The ring metamaterials in initial state from top to bottom are constructed by C_I, C_{II}, C_{III} and R functional groups. The geometry of triangular ring metamaterials is fixed by 3D printed plastic structural supports, see detail in Fig. S27. The outer iron ring and the cords arranged at the evenly divided position are used to apply the external force at outer wall of ring metamaterials. The rendered images in the middle column show the force applied form during deformation. The green arrow indicates functional groups that can be transformed to each other. **b** Radial stretch R_i/R_{i0} varied as a function of sweeping angle ratio $\Delta\theta/\Delta\theta_{max}$. The solid lines represent theoretical results, and the discrete points denote experimental results. **c** Semi-experimental torque-radial strain results of triangular ring metamaterials. The abscissa $\Delta R_i/R_{i0}$ represents the ratio of the variation of the inner radius, as characteristic size, of the metamaterial to its initial inner radius and denotes the difference between deformed radial stretch and initial radial stretch. The negative value corresponds to the torsional contraction deformation, and the positive value corresponds to the torsional expansion deformation. **d** Comparison of equivalent stiffness modulus between triangular rings and those of quadrilateral rings. The equivalent stiffness modulus is calculated by Eq. S44.

The panel c (previous panel d) of Fig. 3 describes the relationship between the change of the characteristic size of the metamaterial and the torque applied to cause this size change. Its horizontal axis $\Delta R_i/R_{i0}$ represents the ratio of the variation of the inner radius, as characteristic size, of the

metamaterial to its initial inner radius. In order to eliminate the influence of different configuration dimensions of the metamaterial on the comparison of deformation quantities, the variations of the inner radius is normalized via dividing them by their initial inner radius. The horizontal axis $\Delta R_i/R_{i0}$ denotes the difference between deformed radial stretch and initial radial stretch. Among them, the negative value corresponds to the torsional contraction deformation, and the positive value corresponds to the torsional expansion deformation. The interpretation and definition content about $\Delta R_i/R_{i0}$ are added in the figure caption.

8. In panel C, do you have a discontinuity around 0?

Response: Thank you for pointing out this question. The panel c (revised panel b) shows that radial strain R_i/R_{i0} varied as a function of sweeping angle ratio $\Delta\theta/\theta_{max}$. In this panel, the negative value represents the torsional contraction, and positive value represents the torsional expansion. The dark red curve and dots represent the data of torsional expansion deformation of C_I ring metamaterial. The C_I ring deforms from initial state at $\Delta\theta/\theta_{max} = 0$ to final deformed state at $\Delta\theta/\theta_{max} = -0.9$. It has continuity around 0. The red curve and dots represent the data of torsional expansion and contraction deformation of C_{II} ring metamaterial. The C_{II} ring can deform from contracted state at $\Delta\theta/\theta_{max} = -0.5$ across undeformed stable state at $\Delta\theta/\theta_{max} = 0$ to expanded state at $\Delta\theta/\theta_{max} = 0.45$, and vice versa. It has continuity around 0. This metamaterial realizes torsional contraction and expansion continuously. The blue curve and dots, magenta curve and dots represent the data of torsional contraction deformation of C_{III} ring metamaterial and R ring metamaterial. The two metamaterial deforms from initial states at $\Delta\theta/\theta_{max} = 0$ to deformed states. They have continuity around 0. From the graph, it can be observed that when the abscissa $\Delta\theta/\theta_{max}$ is equal to 0, the curves relating to the C_I , C_{II} , C_{III} , and R rings overlap at the same point, indicating that their initial geometric topologies are the same. However, it is important to note that, as shown in Fig. 3a, their actual initial configurations in terms of element arrangement are different from each other. Therefore, a reconstruction of functional groups needs to be performed at the position of abscissa = 0 in order to transit to other curves. In other words, there is no direct continuity between the curves, which may be what the reviewer refers to as "discontinuity". For deformations with functional group transformations between C_I and R rings, C_{II} and R rings, C_{III} and R rings, they are discontinuous around 0. We add the explanation about the continuity around 0 on page 9 line 224-228 and as follows.

It is important to note that, a reconstruction of functional groups needs to be performed at the position of $\Delta\theta/\Delta\theta_{max} = 0$ in order for deformation curve transitions between C_I and R rings, C_{II} and R rings, C_{III} and R rings. Consequently, the point of $\Delta\theta/\Delta\theta_{max} = 0$ is discontinuous for deformation involving functional group transformation while it is continuous for deformation of the same type of ring metamaterial.

9. Fig 4 and 5 are also not clear and way to heavy.

Response: We are grateful to the reviewer for these helpful suggestions. We refined the Fig. 4 and 5 in revised manuscript. The Fig. 4 mainly expresses the multidirectional axial deformation of quadrilateral ring metamaterials and their mechanical performance. The quadrilateral ring metamaterial with C_I functional groups can produce expansion under axial tension and expansion of auxetic behaviour. The quadrilateral ring metamaterial with C_{III} and R functional groups can produce contraction under axial compression and contraction of auxetic behaviour. In addition, the quadrilateral ring metamaterial with C_{II} functional groups can produce expansion under axial tension, expansion of auxetic behaviour, contraction under axial compression and contraction of auxetic behaviour. The original figure is not clear and way to heavy because it contains all the metamaterials at initial state and corresponding deformations. The quadrilateral ring metamaterial with C_{II} functional groups can undergo four types of aforementioned deformation. Therefore, we retain the content of quadrilateral ring metamaterial with C_{II} functional groups for more comprehensive deformation demonstration. We have simplified panel a, b, c of fig. 4 as panel a of the revised figure. And the demonstration of quadrilateral ring metamaterial with C_I and C_{III} functional groups are moved to supplementary material. Panel d shows the stress-strain curves of all four metamaterials. Panel e shows the theoretical comparison of equivalent stiffness modulus between C_I , C_{II} , C_{III} and R metamaterials for multidirectional axial deformation. Contents of panel d and e remain unchanged but renamed as panel b and c. The revised Fig. 4 is shown as follows. The corresponding text is revised (page 11, lines 262-269).

Fig. 4 The multidirectional axial deformation and mechanical characterization of quadrilateral ring metamaterials.

a The prototype snapshots of designed quadrilateral rings with C_{II} and R functional groups at initial stable state, the final states with axial deformation with Poisson's ratio $\nu = 0$ and auxetic deformation with $\nu = -1$. The geometry of quadrilateral ring metamaterials is fixed by 3D printed plastic structural supports, see detail in Fig. S27. The green arrows in snapshots point out the forced direction. **b** The stress-strain curves and **c** Comparison of stiffness modulus of the quadrilateral ring constructed by C_I , C_{II} , C_{III} and R functional groups under multidirectional axial deformation. The solid lines and hollow dots in **b** represent theoretical results and experimental results, respectively.

The Fig. 5 mainly expresses the axial deformation of quadrilateral ring metamaterials under two evolutionary morphologies. Similarly, the original figure is not clear and way to heavy because it contains all the four metamaterials at initial state and final deformed state. Thus, we have simplified panels a, b and e of previous Fig. 5 as panels a and d of the revised figure. The quadrilateral ring metamaterial with C_I , C_{II} , C_{III} functional groups under evolutionary morphology I and II have the same axial deformation. However, the quadrilateral ring metamaterial with C_{II} functional groups can undergo two types of aforementioned deformation. Therefore, we retain the content of C_{II} ring metamaterial under evolutionary morphology I and II for more comprehensive deformation demonstration. And the demonstration of C_I and C_{III} ring metamaterials under evolutionary morphology I and II are moved to supplementary material. Contents of panel c, d and f in original Fig. 5 remain unchanged but renamed as panel b, c and e. The corresponding text (page 12-13, lines 282-295) is also revised.

Fig. 5 Two evolutionary morphology of quadrilateral ring metamaterials and their mechanical properties. **a** The prototype snapshots of initial stable state (upper column) and auxetic deformation (lower column) of designed quadrilateral rings with complete-elastic (example: C_{II} ring with $\Gamma=60^\circ$, left) functional groups in evolutionary morphology I. And those of transformed ring metamaterials with rigid-elastic (R, right) functional groups in evolutionary morphology I. **b** The continuous reprogrammable force-displacement curve of quadrilateral ring metamaterial with rigid-elastic functional groups as the variation of adjustable angle Ω . The solid lines represent the theoretical results, while the hollow points represent the experimental results. **c** The continuous reprogramming of stiffness modulus at the initial state for reprogrammable mechanical metamaterial with quadrilateral ring in first evolutionary morphology. **d** The prototype snapshots of initial stable state (upper column) and axial deformation with Poisson's ratio $\nu = 0$ (lower column) of designed quadrilateral rings with complete-elastic (example: C_{II} ring with $\Gamma=0^\circ$, left) in evolutionary morphology II. And those of transformed ring metamaterials with rigid-elastic (R, right) functional groups in evolutionary morphology II. **e** The experimental stress-strain curves of quadrilateral rings constructed by C_I , C_{II} , C_{III} and R functional groups in evolutionary morphology II.

10. Generally here the overall paper should be rewritten, and figure done properly. Math symbols should respect the standards (not bold for scalar, not italic for the names...)

Response: We thank the reviewer for pointing out this. We re-adjusted the layout of the article based on all comments from the reviewers. The content of the figure is refined, and the full text expression is modified. The scalars are formatted in italics but not bold. The italic formatting are cancelled for names in revised manuscript. We accordingly modify the relevant contents in the revised manuscript. Due to numerous revisions made, we kindly refer you to the revised documents for a detailed account of the modifications.

11. Recent literature on triangular and rotating squares should be cited.

Response: Thank you for your suggestion. Recent papers about metamaterial research about triangular and rotating squares are investigated and cited in the discussion about metamaterials deformation and negative Poisson's ratio. The modified content about adding recent triangular literature (page 15 line 342-347) is as follow:

Although the triangular ring metamaterial is difficult to deform under uniaxial force, the metamaterial periodically arranged with triangular ring with a form of planar network is easy to deform under uniaxial tension and compression. The triangular-ring periodic homogeneous metamaterial is considered as a material consisting of randomly packed smooth spherical particles²². Its Poisson's ratio is a constant of -1, related to the stiffness ratio of non-deformable to deformable directions of functional groups³⁵⁻³⁷.

Triangular literature:

35. Mizzi L , & Spaggiari A . Lightweight mechanical metamaterials designed using hierarchical truss elements. *Smart Materials and Structures* **10**, 105036 (2020). DOI:10.1088/1361-665X/aba53c.
36. Rafsanjani A , & Pasini D. Bistable Auxetic Mechanical Metamaterials Inspired by Ancient Geometric Motifs. *Extreme Mechanics Letters* **9**:291-296 (2016). DOI:10.1016/j.eml.2016.09.001.
37. Mengqi Wan, Keqin Yu, & Huiyu Sun, 4D printed programmable auxetic metamaterials with shape memory effects, *Composite Structures* **279**, 114791 (2022). DOI: 10.1016/j.compstruct.2021.114791

The modified content about adding recent literatures (page 13-14 line 316-325) is as follow:

The negative Poisson's ratio (auxetic) effect primarily occurs in stretch-dominated structures where the shear stiffness is greater than the axial stiffness²²⁻²⁵, rigid units with relative rotation²⁶⁻²⁸, chiral lattices with node rotation and ligament bending^{24,29}, and sheet-like origami with reoriented structural elements^{28,30}. The R/C functional group constructed in this study belongs to a stretch-dominated structure and achieves the microstructure concept of a shock absorber element with a negative Poisson's ratio, as described in literature 22. In constructing two-dimensional material networks, this functional group, simplified as a spring, can be more easily arranged in various polygon network structures, such as triangles or quadrilaterals, leading to the fabrication of more diverse material structures and properties, compared to re-entrant structures⁵, rotating cube structures³¹⁻³⁴, and Miura origami⁶.

Rotating squares:

32. Deng B, Yu S, Forte AE, Tournat V, & Bertoldi K. Characterization, stability, and application of domain walls in flexible mechanical metamaterials. *Proceedings of the National Academy of Sciences of the United States of America* **117**(49):31002-31009 (2020). DOI: 10.1073/pnas.2015847117.
33. Sorrentino A., Castagnetti, D., Mizzi., & Spaggiari, A. Rotating squares auxetic metamaterials with improved strain tolerance. *Smart Materials and Structures* **30**(3), 035015 (2021). DOI:10.1088/1361-665X/abde50
34. Choi, G.P.T., Dudte, L.H. & Mahadevan, L. Programming shape using kirigami tessellations. *Nature Materials* **18**, 999–1004 (2019). DOI: 10.1038/s41563-019-0452-y

Reviewers' Comments:

Reviewer #1:

Remarks to the Author:

The authors addressed all my comments and revised the manuscript correctly. The current manuscript ensures the originality of the mechanical metamaterial design the authors developed, which is why I am happy to recommend this article for publication in NCOMMS.

Reviewer #2:

Remarks to the Author:

Authors have properly answered all queries.

RESPONSE TO REVIEWER COMMENTS

Reviewer #1

The authors addressed all my comments and revised the manuscript correctly. The current manuscript ensures the originality of the mechanical metamaterial design the authors developed, which is why I am happy to recommend this article for publication in NCOMMS.

Response: We thank the reviewer for the time invested in making this paper good enough to be accepted.

Reviewer #2:

Authors have properly answered all queries.

Response: We appreciate the reviewer's help.